# Prototype Transformer: Towards Language Model Architectures Interpretable by Design

Yordan Yordanov [* 1]   Matteo Forasassi [* 1]   Bayar Menzat [1]   Ruizhi Wang [1]   Chang Qi [1]   Markus Kaltenberger [1]
Amine M'Charrak [2]   Tommaso Salvatori [1]   Thomas Lukasiewicz [1 2]

## Abstract

While state-of-the-art language models (LMs) surpass the vast majority of humans in certain domains, their reasoning remains largely opaque, reducing trust and risking deception and hallucination. In this work, we introduce the Prototype Transformer (ProtoT)—an autoregressive LM architecture that replaces the quadratic-cost self-attention in the transformer with a linear-cost module based on prototypes (parameter vectors). In ProtoT, the prototypes create communication channels aggregating contextual information at different time scales. We show that this leads to the prototypes automatically capturing nameable concepts (e.g. "woman") during training, and it provides the potential to interpret the model's reasoning and do targeted edits of its behavior. Compared to baselines, ProtoT scales well with model and data size, shows robustness to input perturbations, and performs well on text generation and downstream tasks (GLUE). These results suggest that ProtoT is a promising step toward autoregressive language models that are more interpretable by design.

## 1. Introduction

Large-scale autoregressive language models have achieved strong performance across various domains, with architectures like GPT-4 and LLaMA (Achiam et al., 2023; Touvron et al., 2023a) demonstrating capabilities on benchmarks spanning mathematical reasoning, code generation, and natural language understanding tasks. However, these models exhibit limited transparency in their reasoning processes, creating challenges for understanding how they arrive at their outputs and potentially limiting their deployment in applications where interpretability is important. For example, it has been observed that there is a large disconnect between models' explicit reasoning and their internal computational processes (Greenblatt et al., 2024): while language models can generate step-by-step explanations when prompted, research indicates that these explanations may not reflect their actual reasoning pathways (Turpin et al., 2023). This opacity also contributes to hallucination behaviors, where models generate confident but factually incorrect outputs without clear indicators of uncertainty (Zhang et al., 2025).

Current interpretability methods for language models primarily operate as post-hoc analysis tools on architectures not designed with interpretability as a primary consideration. Approaches such as attention visualization (Clark et al., 2019), probing techniques (Tenney et al., 2019), and causal intervention methods (Meng et al., 2022) provide insights into model behavior but face limitations imposed by the underlying self-attention architecture. More recent techniques like sparse autoencoders (Kissane et al., 2024) attempt to disentangle superposed features within existing architectures, training a secondary model on a model component of interest to decompose its internal representations.

In this work, we present the Prototype Transformer (ProtoT),[1] an alternative autoregressive LM architecture incorporating interpretability considerations directly into its design. ProtoT replaces the standard self-attention with a prototype-based mechanism, where learnable parameter vectors (prototypes) define separate communication channels with the input sequences. This design choice allows prototypes to capture interpretable concepts during training, providing more direct access to the model's reasoning components.

ProtoT offers several characteristics that distinguish it from standard transformer architectures. The prototype-based design enables direct inspection and modification of learned concepts, supporting targeted behavioral adjustments based on identifiable reasoning components with negligible col-

---

[*]Equal contribution  [1]Vienna University of Technology, Austria; [2]University of Oxford, UK. Correspondence to: Yordan Yordanov <yordan.yordanov@tuwien.ac.at>, Matteo Forasassi <matteo.forasassi@tuwien.ac.at>.

*Proceedings of the 43rd International Conference on Machine Learning*, Seoul, South Korea. PMLR 306, 2026. Copyright 2026 by the author(s).

[1]The code can be found at: https://github.com/YDYordanov/prototype_transformer.

lateral damage to the model's sequence perplexity. The architecture aggregates contextual information across different time scales through prototype communication channels, which facilitates the interpretation of both local and global reasoning patterns. Additionally, ProtoT operates with linear computational complexity relative to sequence length, versus quadratic for the standard self-attention. The explicit prototype representations enable attribution of predictions to the internal pathways that generated them, allowing inspection of how information is routed. Our contributions briefly are:

- We introduce ProtoT, a novel autoregressive LM architecture replacing self-attention with prototype-based communication. ProtoT uses learnable parameter vectors for multi-channel communication with input sequences, achieving linear computational complexity and performance competitive with linear baselines.

- We demonstrate that prototypes automatically learn disentangled, nameable concepts during training, across abstraction levels, enabling more interpretable analysis of model reasoning. We quantify and show that these behaviors compare favorably against standard transformers. We demonstrate targeted, highly surgical behavior edits for a wide range of concepts through selective prototype intervention with minimal degradation in sequence perplexity. We also show per-prototype time preference, and "predict and consolidate" behavior patterns.

- We provide extensive evaluation showing that ProtoT achieves competitive performance to linear-compute baselines, while outperforming quadratic-compute self-attention on text generation. The architecture also demonstrates stability under meaning-preserving perturbations, mediated by the prototypes.

## 2. Related Work

**Interpretability in LMs.** One of the main goals when it comes to interpreting language models is to identify which components—such as heads, layers, or neurons—are responsible for specific behaviors (Zhang & Nanda, 2024). This is non-trivial, as attention magnitude does not necessarily imply causal importance (Jain & Wallace, 2019). Moreover, phenomena like *superposition*, where multiple features are encoded in overlapping directions, make isolating concepts difficult (Elhage et al., 2022). Consequently, this problem is often approached via causal intervention, analyzing activation differences between clean and corrupted prompts (Meng et al., 2022; Geva et al., 2023; Wang et al., 2022), with recent work automating the discovery of minimal circuits responsible for specific behaviors (Conmy et al., 2023). Complementary approaches inspect intermediate layer rep-

resentations by projecting them into vocabulary space to trace how predictions form across depth (Nostalgebraist, 2020; Belrose et al., 2023). A promising direction involves Sparse Autoencoders (SAEs), applied not just to MLPs but also to attention outputs (Bricken et al., 2023; Kissane et al., 2024), aiming to disentangle superposed features into interpretable units (Rai et al., 2024). Recent works use LMs to automate interpretability analysis (Paulo et al., 2025).

**Prototype Methods.** Prototype methods seek to render decisions interpretable by relating inputs to learned examples. In computer vision, this often involves comparing inputs to prototypical parts for classification (Chen et al., 2019; Rymarczyk et al., 2022). Recently, ProtoViT (Ma et al., 2024) adapted this to Vision Transformers, using prototypes as deformable parts in the final layer. In NLP, approaches like ProtoAttend (Arik & Pfister, 2020) use attention over entire training examples for decision-making. Other architectures, such as ProtoryNet (Hong et al., 2023) and ProSeNet (Meng et al., 2022), introduce prototype trajectories or sparsity constraints to refine interpretability. Unlike many of these works which place prototypes only at the final classification stage, ProtoT integrates prototypes at every level of hierarchy. Recent advances also include ProtoLens for sub-sentence span extraction (Wei & Zhu, 2025) and white-box frameworks for sentiment detection (Wen & Rezapour, 2025).

**Alternatives to Self-Attention.** Recent work has explored replacing self-attention by using fixed sets of latent vectors. Slot Attention (Locatello et al., 2020) employs a competitive binding mechanism (softmax over slots) to segment inputs, but relies on iterative refinement steps (e.g., GRU) over static inputs. The Perceiver family (Jaegle et al., 2022; Hawthorne et al., 2022) decouples compute from input size by projecting data into a latent space processed by a standard transformer stack. Our ProtoT mixer differs fundamentally in both interaction and state update. Unlike Perceiver, where latents interact globally via self-attention ($O(R^2)$), ProtoT prototypes never interact; they serve as filters for $R$ independent, parallel channels ($O(R)$). And unlike Slot Attention's iterative refinement, ProtoT updates state autoregressively via strict past-only time-discounted aggregation (EMA). This design creates a semantic routing bottleneck rather than a general-purpose processing workspace, encouraging prototypes to capture nameable concepts (Sec. 5.1).

## 3. Prototype Transformer

The prototype transformer (ProtoT) is an autoregressive LM architecture, based on prototypes. It is a transformer stack identical to LLaMA-3 (Grattafiori et al., 2024), apart from the mixer module: ProtoT uses a prototype-based mixer instead of standard self-attention. Like LLaMA-3, ProtoT has L blocks ("layers"), each consisting of a mixer and a SwiGLU (Shazeer, 2020) feed-forward (FFN) module

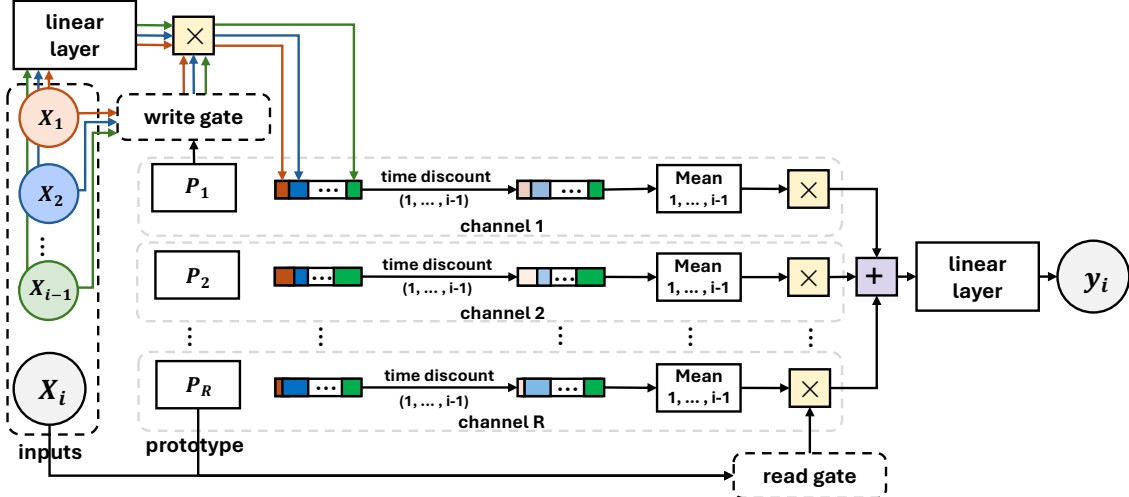

*Figure 1.* A single autoregressive step of the ProtoT mixer module. Prototypes $P_1, \ldots, P_R$ route inputs $x_1, \ldots, x_{i-1}$ (past-only – excluding $x_i$) into $R$ channels via similarity scores at the *write gate*. Time-discount and time-wise mean are applied per channel. The *read gate* reads from each channel via the similarity between its prototype and $x_i$, followed by aggregation into the output $y_i$.

with the same intermediate ratio of $\sim$2.7 as in LLaMA-3, and skip-connections with RMS pre-layernorm (Zhang & Sennrich, 2019) around each mixer module and FFN.

**Prototype mixer:** This module uses $R$ prototypes (trainable parameter vectors) to route the communication across the sequence through R corresponding channels (Fig. 1). Communication in and out of the channel is mediated via a write gate (in) and read gate (out) (Eq. 1 and 2). Each prototype is used as a filter via the write gate to aggregate (time-discounted) information from the past, defining a communication channel, and reading back the information via the read gate into the sequence. It follows this formula applied at any token position $i$, for linear maps U, V, and W:

$$y_i = U\left(\sum_{k=1}^{R}\underbrace{\text{Softmax}_k\left(\frac{W(x_i) \cdot \mathbf{P}_k}{\tau_r}\right)_k}_{\text{Read Gate}} \underbrace{\text{PM}_k}_{\text{Prefix Mean, channel } k}\right), \text{ where:}$$

$$\tag{1}$$

$$\text{PM}_k = \frac{\sum_{j\,:\,j<i} \beta_k^{i-j} \underbrace{\text{Softmax}_k\left(\frac{x_j \cdot \mathbf{P}_k}{\tau_w}\right)_k}_{\text{Write Gate}} V(x_j)}{\underbrace{\sum_{j\,:\,j<i} \beta_k^{i-j} \text{Softmax}_k\left(\frac{x_j \cdot \mathbf{P}_k}{\tau_w}\right)_k}_{\text{Mass Normalization}}}$$

$$\tag{2}$$

Communication passes through an R-channel bottleneck, where each channel is mediated by a prototype via the read and write gates. Since each prototype defines how context is averaged within its channel, multiple meanings assigned to the same channel are forced into a compromise representation, blurring their distinctions. This creates pressure

for semantic specialization across channels. In Sec. 5.1, we show that this effect is reflected at the write gate where prototypes capture nameable, partially disentangled concepts.

**Write gate:** a cross-attention-like gating mechanism between prototypes $P_k$ and inputs $x_j$ (Eq.2), reweighing the values ($V(x_j)$). Unlike cross-attention, the Write Gate uses softmax over the prototypes, to do channel-aligned rather than sequence-aligned gating. It writes information from the sequence into the R channels, and uses a learned temperature $\tau_w$ for added expressivity. At layers 0 and 1 we also add a *local convolution* at the values stream (immediately after $V(x_j)$) – a convolution with kernel size 5, across the 4 past tokens and the current one, with h channels corresponding to the hidden dimension. It adds expressivity at the value stream, by capturing short-term relationships, and we show reduced perplexity and increased utility of layer 0, as measured by increased alpha-gate (A.6.1, Table 16). We also verify the kernel size in Appendix A.6.2, Table 17.

**Read gate:** a cross-attention-like mechanism that reads information from the R channels according to similarity with the corresponding R prototypes Eq.2. It is identical to the write gate, except for the linear map $W(x_i)$ and separate temperature $\tau_r$, which add expressivity and allow for read/write gate specialization. This specialization may also help with interpretability, by decoupling the reading and the writing mechanisms. At layer 0 we use *shared read/write routing* (removing the $W$ linear map) and sharper $\tau_r$ initialization (3.0 vs 1.0) which we show (Appendix A.6.1, Table 16) reduce perplexity and increase utility (higher alpha-gate value, defined below), likely by providing an inductive bias that reduces noise.

**Prefix mean (PM):** aggregator of past information via R communication channels, each corresponding to a prototype. In PM, a cumulative sum operation (Eq. 2) enforces a *strict autoregressive constraint* for next-token prediction. At position $i$, the aggregation is only on entries from positions $j < i$, so the representation used for predicting token $i$ is a function of the past context only. In contrast, standard self-attention allows position $i$ to attend directly to itself, providing a vertical shortcut from the input at position i to the output at the same position. By removing this direct path, the prefix mean encourages the write gate to base its updates on earlier positions and to anticipate the needs of the read gate, which we empirically demonstrate in Section 5.1. A *discounted prefix* gives per-channel time preference, allowing aggregation at different time scales. It is defined as exponential moving average (EMA) (time discount) on the Prefix Mean, via $\beta_k = \sigma(\gamma_k) \in (0,1)$, for learnable $\gamma_k$. It can also be used to interpret the time preference of each prototype, as in Section 5.1. *Mass normalization* then turns the prefix cumulative sum into a prefix mean by dividing it by the sum of coefficients. This theoretically stabilizes the computation, and we have observed reduces perplexity at a small computational cost (Appendix A.6.2, Table 19). We also use *low-rank projection* at $1/2$ of the hidden size (h) at the value stream ($V(x_j)$), which saves up to 50% compute at the mixer module, with similar performance (Table 19). We keep the prototypes and routing (read and write gates) in the full size h as their computational cost is only linear in h.

**Alpha Gate:** a ReZero-like (Bachlechner et al., 2021) scalar gate applied at the output of each Prototype Mixer module before it merges with the residual stream (skip-connection). Unlike ReZero, which uses it to improve training of extremely-deep models, we use it as a low-compute-cost debugging tool: a low value of $\alpha$ at a given layer (declining rapidly during training) is strong evidence that the mixer is not contributing to the final prediction. Due to this role of $\alpha$, we initialize it at identity (1.0) (vs ReZero's 0.0), which also performs better (see Appendix A.6.2, Table 18).

**Compute:** The ProtoT computational cost scales linearly in sequence length, as visible by Eq. 1 and 2. Note the recurrence in Eq. 2: the Prefix Mean for $x_i$ depends only on the Prefix Mean for $x_{i-1}$ and on $x_{i-1}$, both of which can be cached. This means that the model can generate tokens at sequence-wise constant ($O(1)$) compute and memory cost.

## 4. Experimental Setup

**Baselines:** We compare ProtoT to three representative mixer families while keeping the backbone fixed: depth (6), hidden size (256), FFN ratio ($2.7\times$), RMSNorm, dropout (0.1), and the training recipe. We use the same tokenizer and optimizer across models and do not reuse any pre-trained weights.

To isolate mixer effects, we exclude MoE (e.g., Qwen-3) (Yang et al., 2025) and hybrid architectures (e.g., Jamba) (Lenz et al., 2025). We compare against a *LLaMA-style Transformer*: a single-expert, decoder-only self-attention Transformer following LLaMA-3/3.1 (Grattafiori et al., 2024), matched to ProtoT in backbone hyperparameters and training setup, with the only difference in the mixer (self-attention vs. prototype); *Mamba* (Gu & Dao, 2024): a modern state-space model (SSM) instantiation with the same dimensionality (6 layers, 256 hidden size) as ProtoT; and *DeltaNet (Yang et al., 2024a) (delta-rule linear transformer)*: a linear-attention baseline, configured with the same width, depth, and FFN ratio as ProtoT.

**Dataset:** We use a subset of the FineWeb-Edu dataset (Penedo et al., 2024), a high-quality diverse web crawl specifically curated for LM training. Our sampled dataset contains 360,313 documents for a total of 250M tokens, with train/dev/test split of 338,695/18,015/3,603 documents. We use a custom BPE tokenizer (Sennrich et al., 2016) trained on the dataset with a vocabulary size of 16,000 tokens.

**Hyperparameters, scheduler, and optimizer:** We do hyperparameter search on the default training settings: 18k documents for 10 epochs of the training data, with the default model sizes: hidden size h=256, layers L=6, and context ctx=256, unless otherwise specified. We use automatic search over the batch size and learning rate (LR). See Appendix A.1 for more details. We use linear warmup over 2% of training, and cosine annealing towards 10% of the peak LR. We train all our models with AdamW (Loshchilov & Hutter, 2019), following standard practice for LMs, and because AdamW is more robust to hyperparameter choice (Zhao et al., 2025) than SGD.

**Dropout:** For all models, we use dropout (with probability 0.1) after the token embeddings, at the residual (block output) between blocks, and inside the FFN, because we find that it reduces perplexity for all models (Appendix A.6.2, Table 21). This is likely because it prevents overfitting in the multi-epoch training regime (10 epochs) that we use. For LLaMA, we additionally put dropout inside the self-attention, which further decreases perplexity.

**Attention heads and prototypes (R):** Similar to (Press & Wolf, 2017), we have found that sharing the weights between embeddings and LM head reduces perplexity at the hyperparameter search stage, for all models. This is likely because it provides an inductive bias aligning the token embeddings between input and final projection. We also do this for large-scale experiments, for simplicity. We select the attention heads from $\{2, 4, 8\}$, but at both small-scale and large-scale runs we have found that 4 works best for all models with attention heads (LLaMA and DeltaNet), which

is what we use. For ProtoT's prototypes (R), we have found diminishing returns in terms of perplexity improvements beyond R=32 (Appendix A.6.2, Table 20), while computation scales linearly with R. Therefore, we use R=32 for all runs.

# 5. Experiments

*Table 1.* Scalability: *Columns 1–4:* context sizes from Default=256 to 2048 at default settings (h=256, L=6, data=18k). *Column 5:* L-scale (Large-scale) settings (h=512, L=12, ctx=512, data=339k). Test perplexity (lower is better). The best results are in bold.

| Model | Default | 512 | 1024 | 2048 | L-scale |
|---|---|---|---|---|---|
| LLaMA | **78.7** | **67.8** | **62.7** | **61.9** | **25.8** |
| Mamba | 86.0 | 74.6 | 69.3 | 67.8 | 26.5 |
| DeltaNet | 90.4 | 74.6 | 68.1 | 67.5 | 31.5 |
| ProtoT | 90.5 | 84.8 | 80.7 | 83.0 | 29.5 |
| ProtoT (h=512) | 91.6 | **76.7** | **70.3** | **70.2** | – |
| ProtoT (L=12) | **88.8** | 78.0 | 71.6 | 71.8 | – |
| ProtoT (R=64) | 91.0 | 80.7 | 76.3 | 76.1 | – |

*Table 2.* Evaluation of LM text-generation quality. Elo scores are derived from pairwise judge evaluations (higher is better). The best result is in bold.

| Model | Elo |
|---|---|
| LLaMA | 975.18 |
| Mamba | **1041.79** |
| DeltaNet | 961.80 |
| ProtoT | 1021.24 |

**Large-scale training:** In Table 1, we compare ProtoT to the 3 baselines at large-scale training (first vs. last column). We study the effect of simultaneously scaling the hidden size 2x, the layers 2x, the context size by 2x, and the training data ∼19x, versus the default training settings. The results show that ProtoT scales well to the large model/data scenario. We show that ProtoT maintains relative performance to LLaMA, or even improves it (15.0 → 14.3% worse) with scale. Furthermore, ProtoT outperforms the DeltaNet linear-attention baseline (29.5 vs. 31.5 perplexity, respectively). However, a large gap remains versus LLaMA and the Mamba state-space model (29.5 vs. 25.8 and vs. 26.5, respectively). While we did our best to optimise ProtoT, this is the first iteration of the model, whereas established LMs like LLaMA have had multiple (Touvron et al., 2023a;b; Grattafiori et al., 2024). We expect with community feedback and further refinement to shrink this gap.

**Long-context scalability:** The results in Table 1 (columns 1-4) show that ProtoT scales poorly with context length (if other model dimensions are fixed), which suggests that ProtoT is running into a bottleneck. This is likely because the cross-sequence communications pass through the prefix

mean (Fig. 1 and Eq. 1), over R channels with h hidden dimensions each, which can be restrictive. We further investigate this issue in the final 3 rows of Table 1, where we compare possible culprits: the hidden size h, the number of prototypes R, and the layers L (which can also play a role). The results show that the hidden dimension is the most restrictive as increasing it shows the best context scalability out of the 3 interventions. Our model is most affected by this likely because of our choice to project down to $h/2$ at the values ($V(x_j)$ in Eq.1) to save compute, further exacerbating this bottleneck. In practice, this is less of an issue because, in more realistic settings (e.g. *Large-Scale Training*), the larger capacity of the model would allow for larger context lengths.

**Text-Generation Performance:** To evaluate the quality of outputs, we evaluate open-ended text generation of the large-scale models using an LLM-as-a-judge protocol, following the Chatbot-Arena style pairwise comparison setup. For each prompt, two model outputs are evaluated by a frozen judge model under a fixed rubric, providing win/tie statistics that are converted into Elo scores. As shown in Table 2, ProtoT outperforms LLaMA and DeltaNet (1021 vs 975 and 962 Elo), and is close to Mamba (1042 Elo) in ranking. See Appendix B for text generations from each model.

**Downstream performance:** To comprehensively evaluate the general-purpose language understanding of ProtoT vs baselines, we fine-tune the large-scale models on the GLUE benchmark (Wang et al., 2018) consisting of 9 English NLU tasks spanning sentence- and sentence-pair classification as well as semantic textual similarity (more details in Appendix A.3). As shown in Table 3, LLaMA achieves the best overall performance, but ProtoT remains highly competitive and often matches or outperforms the dense baselines. ProtoT consistently attains the second-best scores on MNLI and MNLI-MM, indicating strong cross-domain robustness for large-scale natural language inference. On single-sentence and sentence-pair classification tasks such as SST-2, QQP and QNLI, ProtoT performs close to LLaMA and on par with Mamba while clearly outperforming DeltaNet, showing that its structured prototype representations do not sacrifice accuracy on high-resource benchmarks. For low-resource tasks such as RTE and CoLA, ProtoT delivers performance comparable to dense models, suggesting that its inductive bias can maintain stable accuracy even when training data is limited. Taken together, these test-set results confirm that ProtoT preserves competitive GLUE performance while offering structural advantages, especially for robust inference under distribution shifts.

**Throughput Benchmarks:** In Appendix A.5, we evaluate both training and inference throughput, and FLOPs. For training, we use identical conditions across models (same

*Table 3.* GLUE downstream fine-tuning results (all metrics reported as percentages). For COLA we report Matthews correlation; for SST-2 accuracy; for MRPC F1; for STS-B Pearson correlation; for RTE, WNLI, QNLI, MNLI and MNLI-MM accuracy; for QQP F1. GLUE reports the unweighted average of the nine task-specific scores. Results are averaged over 3 seeds. Best results are in bold.

| Model | COLA | SST-2 | MRPC | STS-B | RTE | WNLI | QQP | QNLI | MNLI | MNLI-MM | GLUE |
|---|---|---|---|---|---|---|---|---|---|---|---|
| LLaMA | **31.5** | **90.8** | **82.7** | **78.3** | **57.8** | **65.1** | **68.0** | **86.0** | **79.8** | **79.6** | **71.6** |
| Mamba | 31.1 | 88.6 | 80.3 | 72.8 | 54.4 | **65.1** | 64.8 | 82.4 | 74.7 | 74.7 | 68.6 |
| DeltaNet | 13.8 | 85.8 | 80.1 | 67.0 | 50.9 | **65.1** | 62.6 | 80.1 | 71.1 | 71.8 | 64.5 |
| ProtoT | 27.7 | 90.0 | 80.1 | 66.2 | 53.9 | 64.6 | 64.8 | 81.8 | 75.3 | 74.8 | 67.6 |

data pipeline, optimizer, BF16 precision, sequence length 256, batch sizes 32 and 128). ProtoT achieves 25.2 and 7.6 it/s (batch 32/128), outperforming Mamba (11.9 and 3.2 it/s) and DeltaNet (3.5 and 1.8 it/s). However, the LLaMA attention baseline is by far the fastest (55.1 and 23.6 it/s), while having the most FLOPs, which shows its superior optimization in PyTorch. For autoregressive inference with batch size 1, LLaMA has the highest throughput at short context lengths, whereas ProtoT scales better with context and surpasses LLaMA at 32k tokens and beyond, due to its linear-compute architecture; DeltaNet maintains the highest throughput at long context lengths.

**Summary:** While ProtoT trails behind LLaMA in terms of perplexity and downstream performance, ProtoT outperforms LLaMA on text generation. Furthermore, LLaMA's computation cost scales quadratically in sequence length, whereas ProtoT scales linearly. This means the two models are not really in the same class, and the linear baselines (Mamba and DeltaNet) are more suitable to compare against. And indeed, the results show that ProtoT's performance is consistently between that of Mamba and DeltaNet.

### 5.1. Interpretability

Prototypes act as representational slots: contextual information is aggregated into $R$ prototype channels via the write gate and then read back through the read gate (Fig. 1). This structure allows features to be stored and reused within each sequence through the prototype-specific prefix means, enabling association of prototypes with identifiable concepts. Each prototype also has an associated decay parameter $\beta_k$, applied in the prefix mean (Eq. 2) to discount past activations. Smaller $\beta_k$ values produce faster decay, while larger values allow information to persist longer. For interpretability, we report the derived half-life $t_{1/2}^{(k)} = -\frac{\ln 2}{\ln(\beta_k)}$, specifying the expected number of steps for the contribution of prototypes to halve and providing a direct way to analyze specialization in short- or long-term dependencies. We analyze read–write interactions to understand how the model integrates and updates contextual information through prototype channels during sequence generation.

**Experiments:** To investigate interpretability properties, we design four experiments with ProtoT (large-scale model

from Table 1). We compute write routing activations across sequences from the FineWeb validation set for each prototype, aggregate them at the sequence level, and rank sequences by total activation strength. This identifies sequences that most strongly activate each prototype and allows to visually inspect learned concepts and the relation between temporal locality and $\beta_k$ parameter. We also use collected activations to compute widely adopted metrics (L1 sparsity, Gini Coefficient, Entropy, Mutual Information).

We analyze write and read phases during sequence generation. For a subset of prototypes, we select the most activating sequences and compute write and read routing activations for each token along the same prototype. This enables inspection of the internal dynamics of ProtoT, showing how sequence level information is aggregated and maintained during processing.

To quantitatively assess concept disentanglement and polysemanticity, we introduce an extensive analysis and labeling method inspired by auto-interpretability score metrics (Bricken et al., 2023; Paulo et al., 2025). For each prototype, we collect its ten most activating sequences and extract, within each sequence, the tokens with the highest activation. This compact summary of prototype usage is submitted to an LLM-based evaluator (GPT-5.1), which is prompted to produce 1) **Theme:** the main recurring theme identified across the most activating sentences; 2) **Scores (1–10):** a) **Disentanglement:** how clearly the main theme is separated from other themes; b) **Main topic coverage:** how many of the most activating sentences contain the main theme; and c) **Number of themes:** the count of distinct, uncorrelated themes present in the most activating sentences (if the themes are at least 10, assign 10); 3) **Explanation:** a natural-language description explaining the labeling and scoring process. In order to compare the potential for interpretability in ProtoT and Transformer models, we collect activations, compute the same metrics and perform the LLM-aided evaluation experiment also on the trained LLaMA model from Table 1, both by directly inspecting the attention heads and by training a SAE model to recover more interpretable and disentangled feature.

We probe the functional role of individual prototypes through a targeted intervention experiment. Based on write-gate activations on the FineWeb validation set, we identified

*Table 4.* Comparison of LLM interpretability metrics across methods. Disentanglement and Coverage are scored on a 1–10 scale; Num. Themes counts the number of distinct themes identified per feature (lower is better). Best results are in bold.

| Method | Disentanglement ↑ | Coverage ↑ | Num. Themes ↓ |
|---|---|---|---|
| ProtoT | **6.52 ± 1.93** | **7.88 ± 2.25** | **3.86 ± 1.94** |
| LLaMA SAE – Top Variance | 5.91 ± 1.92 | 7.86 ± 1.84 | 4.33 ± 1.82 |
| LLaMA SAE – Top Frequency | 5.52 ± 1.81 | 7.47 ± 1.88 | 4.68 ± 1.82 |
| LLaMA Attention Heads | 5.02 ± 1.28 | 6.69 ± 1.84 | 5.02 ± 1.55 |
| Null Model | 3.20 ± 1.02 | 4.03 ± 1.78 | 6.97 ± 1.28 |

three functionally distinct prototypes from Layer 9: *L9 P7*, which encodes a 'female' concept; *L9 P18*, which partially encodes a 'male' concept; and *L9 P2*, a gender-neutral control. Our intervention consists of disrupting each of these prototypes via parameter re-initialization and measuring the subsequent change in the conditional probability of the target words 'women' and 'girls'. We illustrate these prototypes in Figures 3, 4, and 5. Additional details on the construction of test sentences are in Appendix A.2.1.

**Interpreting prototypes at the write gate:** Results for the LLM-aided interpretability experiment in table 4 show that ProtoT outperforms both direct inspection of LLaMA attention heads and SAE extracted features across all scoring dimensions, with higher disentanglement and coverage and a lower number of uncorrelated topics. These results show that gate-mediated communication at the write gate forms prototypes that can largely be treated as separate, disentangled concept hubs, highlighting their potential for interpretability. Furthermore, human evaluation and LLM-aided scoring reveal that prototypes capture disentangled concepts across varying levels of semantic abstraction. We identify concepts like entity names, functional words, verbs, as well as composite dates, illnesses, or school-related narratives. We also find that these concepts generally reflect the hierarchical organization of the model, with early layers tending to encode more superficial patterns and deeper layers representing composite and abstract semantics. Finally, we identify a correlation between half-life values and encoded concepts, where lower half-life values tend to correspond to local elements (such as stop words, or punctuation). Examples of prototype visualization can be found in Figure 2. Details about the experiment setup for SAE and Null baseline are in Appendix A.8 along with additional prototype examples[2] and the statistics for LLM-aided interpretability experiments for all the methods and for multiple model configurations of ProtoT. The exact prompt used is in Appendix A.9. Interpretability metrics for LLaMA and ProtoT are in Appendix A.7, Figures 7 and 8. In Appendix A.7, we also present the experimental results showing correlation between half-life values and locality.

**Results of the write-read alternation pattern:** We observe

---

²An interactive html for prototypes is in the [code repository](#).

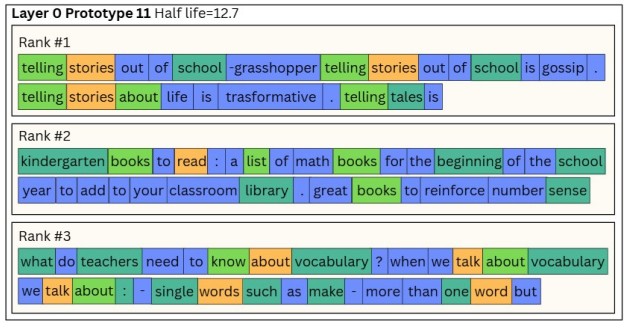

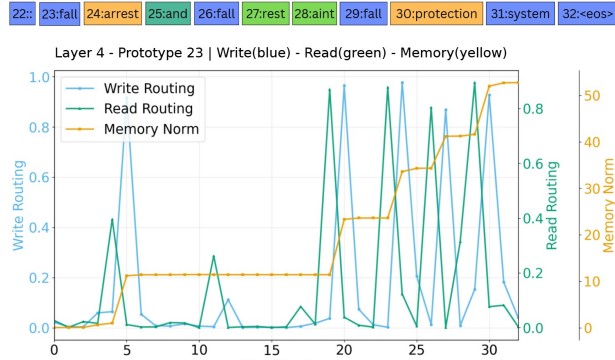

*Figure 2.* *Up*: Sequences most strongly activating prototype 11 at layer 0, which encodes the concept of narrative in a scholastic context. *Down*: Write-gate, read-gate, and memory curves for a sequence that strongly activates prototype 23 at layer 4. Read-gate peaks precede write-gate activations, spiking on the token immediately before those that trigger write-gate routing.

a consistent temporal pattern in read and write activations, with read activity peaking one step before write activity. For example, in the results in Figure 2 (down), write gate activates prototype 4 for the token 'protection' and read gate activates at the preceding token 'fall'. This pattern is consistently seen across the most strongly activating sequences for each prototype and suggests that read and write gates may develop coordinated interactions. This coordination is consistent with a predict-and-consolidate behavior, where the read gate anticipates which prototype may be relevant for the upcoming tokens, and the write gate subsequently updates the memory based on the current token.

*Table 5.* Summary of WriteMask prototype interventions across concepts. For concepts with multiple tested prototypes/seeds, the Max/Mean $\Delta Prob$ is calculated using the most negative sentence-level probability change.

| Concept | Max $\Delta Prob$ | Mean $\Delta Prob$ | Max $\Delta PPL$ | Mean $\Delta PPL$ |
|---|---|---|---|---|
| women | $-16.60\%$ | $-3.13\%$ | $+0.29\%$ | $-0.08\%$ |
| girls | $-10.67\%$ | $-2.36\%$ | $+0.29\%$ | $-0.18\%$ |
| COVID | $-21.97\%$ | $-4.52\%$ | $+5.58\%$ | $+0.76\%$ |
| New Zealand | $-21.54\%$ | $-9.96\%$ | $+3.47\%$ | $+1.62\%$ |
| mental | $-2.20\%$ | $-0.73\%$ | $-0.04\%$ | $-1.20\%$ |

**Results of the prototype intervention:** Our intervention experiments demonstrate that prototypes function as specific and interacting semantic hubs. By employing gate masking (ablating the write/read channels) alongside random re-initialization, we isolated precise mechanistic roles. Disrupting the 'female' prototype $L9\ P7$ significantly decreased the probability of related words (e.g., $-16.60\%$ for 'women'), highlighting its functional importance. In contrast, the control $L9\ P2$ had negligible impact, while disrupting the 'male' prototype $L9\ P18$ increased the probability of female-coded words (e.g., $+16.95\%$ for 'women'). These findings indicate that the model learns functionally distinct prototypes and uses them interactively to refine its predictions.

Table 5 summarizes the same intervention effects across a border set of concepts. We report the maximum and average probability changes after intervention, where we filter out cases where the target word has baseline probability below 1% for reliability. As shown in the results, the model learns functionally distinct prototypes extending beyond gender concepts to geographic entities (e.g., New Zealand') and abstract states (e.g., Mental Health'). Furthermore, cross-seeds experiments on 'COVID' confirm that the emergence of these concept-specific slots is a robust architectural property. Crucially, across all these diverse domains and seeds, the collateral damage of our interventions remains negligible, with perplexity changes remaining within ±1% in most cases and rarely exceeding 4%, suggesting that ProtoT allows for highly surgical behavior steering without destabilizing the model's performance. Full results are in Appendix A.2.

### 5.2. Robustness

We analyze robustness of ProtoT vs baselines (the large-scale models from Table 1) from three complementary perspectives: (1) robustness to *meaning-preserving noise perturbations*, (2) robustness to *prototype clamping perturbations*, and (3) behavior to *interventions* that alter semantics. This unified view clarifies both stability under benign variations and sensitivity to intended changes.[3]

---

[3]The perturbation dataset with generation/filtration scripts, and the manually-generated intervention dataset are in the code repo.

**Noise perturbations:** We first consider *black-box, surface-level perturbations* that preserve meaning (e.g., synonyms, typos, contractions). The perturbation benchmark (Appendix A.4) contains 3,500 semantically equivalent sentence pairs across seven categories. Robustness is quantified by the Jensen–Shannon divergence $JS(p(\cdot|x), p(\cdot|x'))$ between next-token distributions for an original input $x$ and its perturbed variant $x'$. Lower values indicate greater stability. Table 6a shows that Mamba has the overall lowest $JS$, hence the strongest stability. ProtoT, however, consistently outperforms LLaMA on synonyms, typos, spelling, and morphology. This aligns with ProtoT's design: prototypes aggregate contextual information into nameable concepts, yielding stability under lexical variation. While ProtoT lags LLaMA on punctuation (where precise attention alignment is beneficial), it reliably surpasses DeltaNet and is overall competitive with strong baselines.

**Prototype clamping:** To test whether robustness is mediated by prototype routing, we compute *Prototype-Mediated Robustness (PMR)*. For a pair $(x, x')$, let $JS_{\text{base}} = JS(p(\cdot|x), p(\cdot|x'))$. We then clamp the prototype routing weights from $x$ onto $x'$ and recompute $JS_{\text{clamped}} = JS(p(\cdot|x), p^{\text{clamped}}(\cdot|x'))$. We define $PMR = (JS_{\text{base}} - JS_{\text{clamped}})/JS_{\text{base}}$. A positive $PMR$ indicates that prototypes mediate robustness, while negative values suggest residual pathways dominate. Table 6b shows that while the mean $PMR$ is sometimes slightly negative, for 5 out of 7 slices the positive fraction $PMR_{>0}$ is around 0.5–0.6 and $JS_{\text{clamped}} < JS_{\text{base}}$. This shows that prototypes overall contribute to robustness, providing interpretable routing pathways rather than opaque head-level aggregation.

**Intervention behavior:** Finally, we study sensitivity under *interventions* that alter semantics: gender, negation, and number tags. Unlike surface perturbations, these flips should change predictions. We measure $JS$, top-$k$ overlap (Ov), Spearman correlation (Sp), and top-1 invariance (T1). Higher $JS$ and lower Ov/Sp/T1 indicate greater sensitivity to the intervention. Table 6c shows that while DeltaNet attains the highest raw $JS$, ProtoT consistently yields lower Ov, Sp, and T1 compared to LLaMA and Mamba. This indicates that ProtoT adapts more reliably under meaning-altering interventions, reflecting appropriate semantic sensitivity through prototype routing. LLaMA and Mamba often remain insensitive to such tags.

In conclusion, noise perturbation results establish that ProtoT is robust to lexical variation. PMR results show that prototypes actively mediate robustness, exposing interpretable mechanisms. Intervention behavior confirms that ProtoT is more sensitive to meaning-altering changes than Mamba or LLaMA for example. Together, these findings show that ProtoT not only matches or surpasses baselines in robustness but also provides transparent pathways for analyzing

*Table 6.* Robustness results: (a) slice-level JSD; (b) PMR statistics; (c) intervention. Abbreviations: *abbr.*=abbreviation, *contr.*=contraction, *morp.*=morphology, *punct.*=punctuation, *spell.*=spelling, *syn.*=synonym, *typo*=typos. Best values in each column are in bold.

*(a)* Slice-level robustness measured by Jensen–Shannon divergence (lower is better).

| Model | abbr. | contr. | morp. | punct. | spell. | syn. | typo |
|---|---|---|---|---|---|---|---|
| DeltaNet | 1.066 | 0.831 | 0.667 | 0.580 | 0.355 | 0.636 | 0.626 |
| LLaMA | 0.333 | 0.045 | 0.227 | **0.174** | 0.063 | 0.145 | 0.227 |
| Mamba | **0.144** | **0.010** | **0.048** | 0.443 | **0.005** | **0.013** | **0.076** |
| ProtoT | 0.417 | 0.082 | 0.050 | 0.398 | 0.026 | 0.113 | 0.207 |

*(b)* Prototype-Mediated Robustness (PMR). Mean and std of PMR, fraction of positive cases, and average JSDs.

| Slice | $PMR_{mean}$ | $PMR_{std}$ | $PMR_{>0}$ | $JS_{base}$ | $JS_{clamped}$ | n |
|---|---|---|---|---|---|---|
| abbr. | -0.093 | 0.367 | 0.596 | 0.417 | 0.415 | 500 |
| contr. | -0.027 | 0.104 | 0.330 | 0.082 | 0.083 | 500 |
| morph. | -0.034 | 0.176 | 0.474 | 0.050 | 0.051 | 500 |
| punct. | -0.000 | 0.373 | 0.554 | 0.398 | 0.322 | 500 |
| spell. | -0.033 | 0.225 | **0.610** | **0.026** | **0.025** | 500 |
| syn. | **0.013** | **0.075** | 0.606 | 0.113 | 0.109 | 500 |
| typo | 0.001 | 0.279 | 0.533 | 0.208 | 0.186 | 500 |

*(c)* Intervention robustness on gender (gen), negation (neg), and number (num). Metrics: JS (higher better), Ov/Sp/T1 (lower better).

| Model | JS (gen / neg / num) | Ov (gen / neg / num) | Sp (gen / neg / num) | T1 (gen / neg / num) |
|---|---|---|---|---|
| DeltaNet | **0.054 / 0.173 / 0.282** | 0.754 / **0.540 / 0.474** | 0.610 / **0.176 / 0.033** | **0.616 / 0.388 / 0.330** |
| LLaMA | 0.004 / 0.028 / 0.022 | 0.946 / 0.875 / 0.843 | 0.966 / 0.815 / 0.824 | 0.890 / 0.770 / 0.930 |
| Mamba | 0.003 / 0.006 / 0.007 | 0.936 / 0.935 / 0.907 | 0.949 / 0.910 / 0.907 | 0.884 / 0.992 / 0.948 |
| ProtoT | 0.037 / 0.081 / 0.083 | **0.709** / 0.774 / 0.657 | **0.429** / 0.536 / 0.441 | 0.690 / 0.806 / 0.806 |

where robustness arises.

# 6. Conclusion

We have introduced the Prototype Transformer (ProtoT), an alternative autoregressive language model architecture that replaces transformer self-attention with prototype-based communication. ProtoT uses learnable prototypes as multi-channel pathways between input sequences and internal representations, enabling linear computational complexity with only small compromise in performance. It exhibits better text generation than most baselines, downstream performance (GLUE) on par with linear-compute baselines, and strong robustness to perturbations, while scaling well to large model and data size. Beyond performance, the prototypes automatically learn coherent, nameable concepts across abstraction levels and enable targeted, highly surgical behavioral edits with negligible perplexity degradation.

**Limitations:** Prototype interpretability should not be over-trusted – while the LLM-scoring reveals high disentanglement for most prototypes, some remain polysemantic, and prototype labels should be treated as approximations rather than ground truth. However, features identified by post-hoc methods on Transformers have similar risks of being noisy and over-interpreted. In terms of performance, while ProtoT is between Mamba and DeltaNet, it generally underperforms LLaMA. However, ProtoT's compute cost scales linearly in context size, unlike LLaMA's quadratic cost.

Future work may explore the full scope and boundaries of this approach, including broader evaluation across diverse tasks and model scales. Another promising direction is to develop methods for adjusting or aligning prototypes in a pre-trained ProtoT toward semantic roles or concepts that stakeholders care about. This could make prototype-based interpretations more useful for understanding and adjusting model behavior in application contexts where such concepts are central.

In summary, ProtoT shows that incorporating interpretability into architecture design may be compatible with competitive performance. This work contributes to ongoing research toward developing LMs that balance capability with transparency for applications where understanding and correcting model reasoning is essential.

## Acknowledgments

This research was funded in part by the Austrian Science Fund (FWF) 10.55776/COE12 and the AXA Research Fund. Amine M'Charrak gratefully acknowledges support from the Evangelisches Studienwerk e.V. Villigst through a doctoral fellowship.

## Impact Statement

This paper introduces ProtoT as an alternative to self-attention that provides the potential for better interpretability and targeted editability, while performing close to self-attention-level of performance. As such, the societal consequences are mostly positive: enabling debiasing and editing out of undesirable behavior, while providing interpretability mechanisms to potentially detect deception and increase safety. However, an open-weights release of a strong version of ProtoT would also inadvertently have those targeted editability tools exposed, which may provide an easier way for bad (and good) actors to modify the model's behavior.

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

# A. Appendix

## A.1. Additional Details on Experiment Setup for Long-Context Scalability and Large-Scale Training

We use automatic search over batch size (32, 64, 128) and learning rate (from interval (3e-5, 3e-2)). For the search, we use Optuna with BoTorchSampler, with 15-trial warmup and 50 total trials, averaging over 3 seeds per trial. We also use 3 seeds for the long-context scalability results for all models and for the h, L, and R scalability results for ProtoT.

**Batch size:** We found that batch size of 32 works best for training among 32, 64, 128, for all models. Smaller batch sizes were not considered to preserve parallelizability and reduce number of training steps. We keep this batch size (32) in larger experiments as well, for simplicity, and only select the learning rate from a handful of scaling options. Furthermore, smaller batch sizes generalize better than large batch sizes even with large-scale data (Masters & Luschi, 2018); large batch sizes are mainly used for hardware utilization and training speed-up as they require fewer steps to finish training (Ying et al., 2018).

**Learning rate:** The best learning rates found via the automatic hyperparameter search for the default model sizes are: LLaMA: 1.6e-3, Mamba: 3.8e-3, DeltaNet: 6.8e-3, and ProtoT: 2.0e-3.

For the *long-context scalability experiment*, we have tried increasing the learning rate accordingly (by square root of context size ratio), as per AdamW scaling laws (Li et al., 2024), because extended context is computationally-similar to a larger batch size. However, we have found that scaling the learning rate helps only for DeltaNet and only in the large-scale model/data setting. In the results, we report only the best value from scaled vs non-scaled LR for all models.

For the *large-scale training experiment*, we ran each model with the best hyp-s from the hyp search, and with scaled version thereof. We observed instability with Mamba, so we reduced the LR until it reached stability (from 3.8e-3 down to 2.3e-3). For all other models, we report results with the best-found learning rates (above).

## A.2. Prototype Intervention Experiments

To move beyond correlational observations, we designed an intervention experiment to probe the functional role of individual prototypes within the model's predictive process. This methodology involves systematically manipulating a single prototype by either re-initializing it with random noise or zeroing-out the output of the write/read gate corresponding to the prototype, which is equivalent to zeroing-out/ablating the entire communication channel corresponding to it. We then measure the resulting impact on the model's output probabilities for a targeted linguistic task. By quantifying this change, we can assess the prototype's influence and determine its functional importance for a specific prediction.

### A.2.1. IDENTIFYING AND TARGETING CONCEPT-SPECIFIC PROTOTYPES

To identify prototypes that appear to encode distinct, human-understandable concepts, we analyze the top-activating sentences for each prototype from the visualization introduced in Sec. 5.1. Based on this analysis, we selected three prototypes from Layer 9 for our study. The prototype *L9 P7*(Fig. 3), which consistently activates on sentences containing words such as *'women'* and *'girls'*, we hypothesize that *L9 P7 is a key causal component in the model's representation of the 'female' concept*. Similarly, we identified prototype *L9 P18*(Fig. 4) as a representation for the 'male' concept, as it shows high activation for words like *'man'* and *'boy'*. Finally, prototype *L9 P2*(Fig. 5) was selected as a control, as it did not exhibit a clear, gender-coded semantic preference.

**Test Case Construction.** To create a controlled and relevant test set, we began with a seed sentence identified from our initial visualization analysis. This sentence was the top-ranked example from the FineWeb test set that maximally activated the 'write' gate of our primary target, prototype L9 P7. To expand our test set while maintaining semantic consistency, we then prompted a large language model (Gemini 2.5 Pro) to generate six additional sentences thematically similar to the seed sentence, each required to contain the keywords 'women' and 'girls'.

The resulting corpus of seven sentences used in our experiments is as follows:

- *"did you know that there is a government strategy for women and girls in sports and active recreation to address the inequalities of girls' and women's"* (seed sentence from FineWeb)

- *"Many organizations are working on programs that focus on empowering women and girls to participate equally in science and technology."*

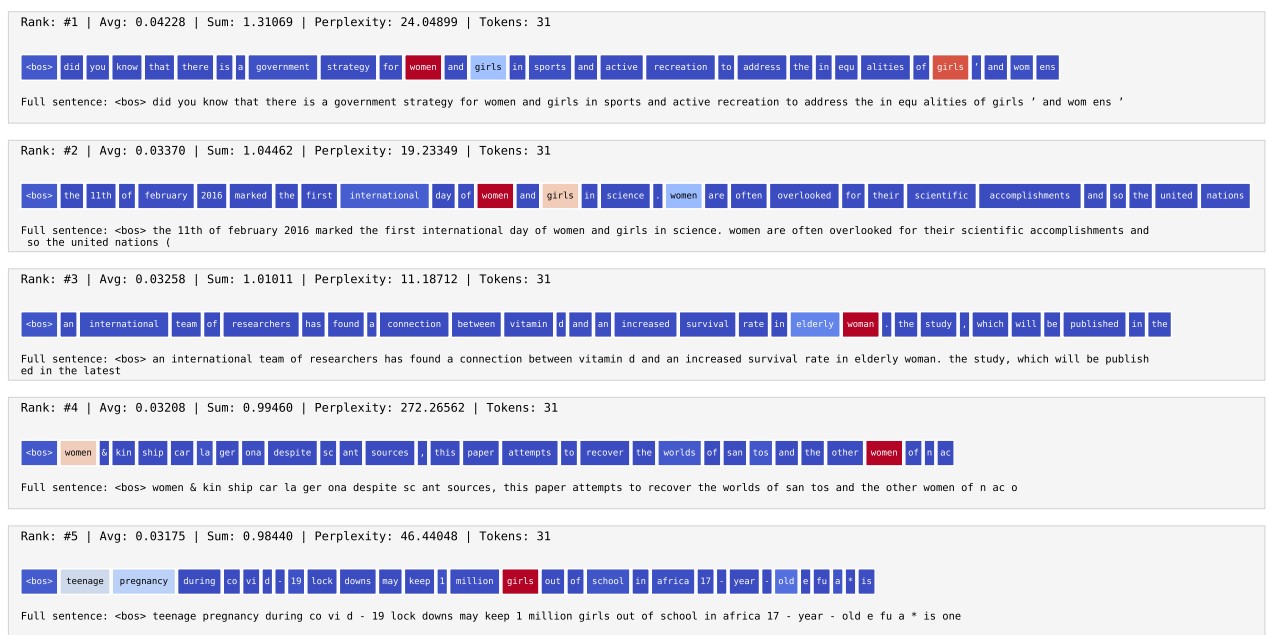

*Figure 3.* Visualization for prototype **L9 P7**

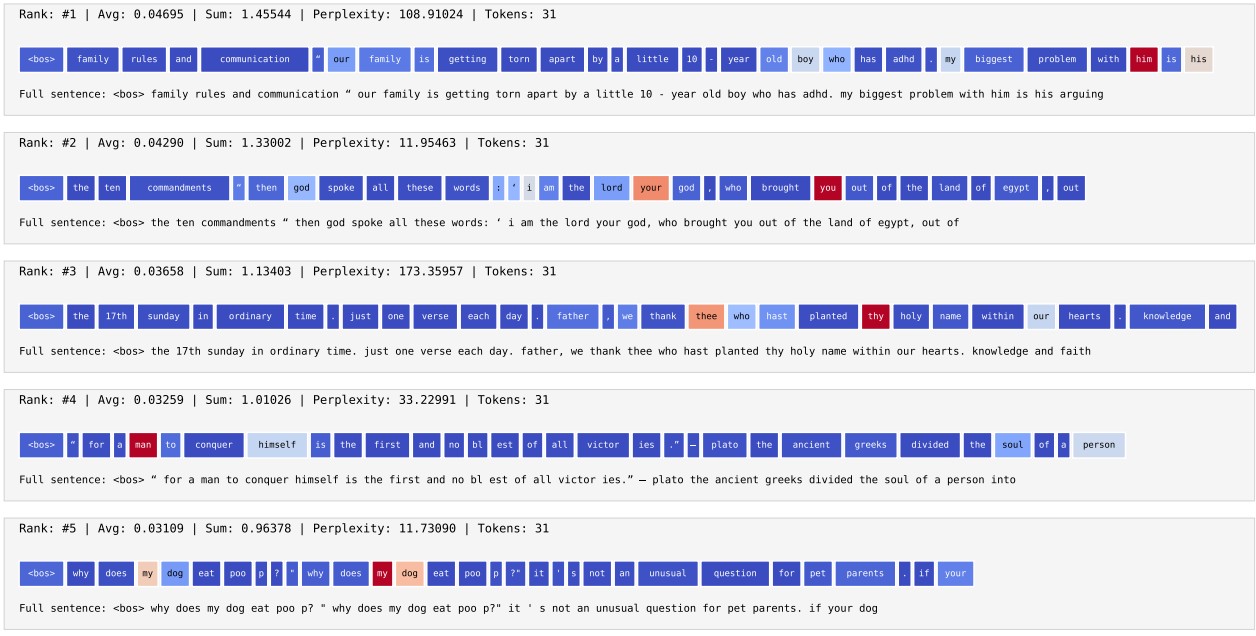

*Figure 4.* Visualization for prototype **L9 P18**

- *"Did you know that several global initiatives aim to protect the rights of women and girls from violence and discrimination?"*

- *"Education policies are increasingly emphasizing equal opportunities for women and girls to excel in leadership roles."*

- *"Access to healthcare remains a critical issue, and governments are creating strategies to improve services for women and girls."*

- *"International campaigns highlight how climate change disproportionately affects women and girls in vulnerable*

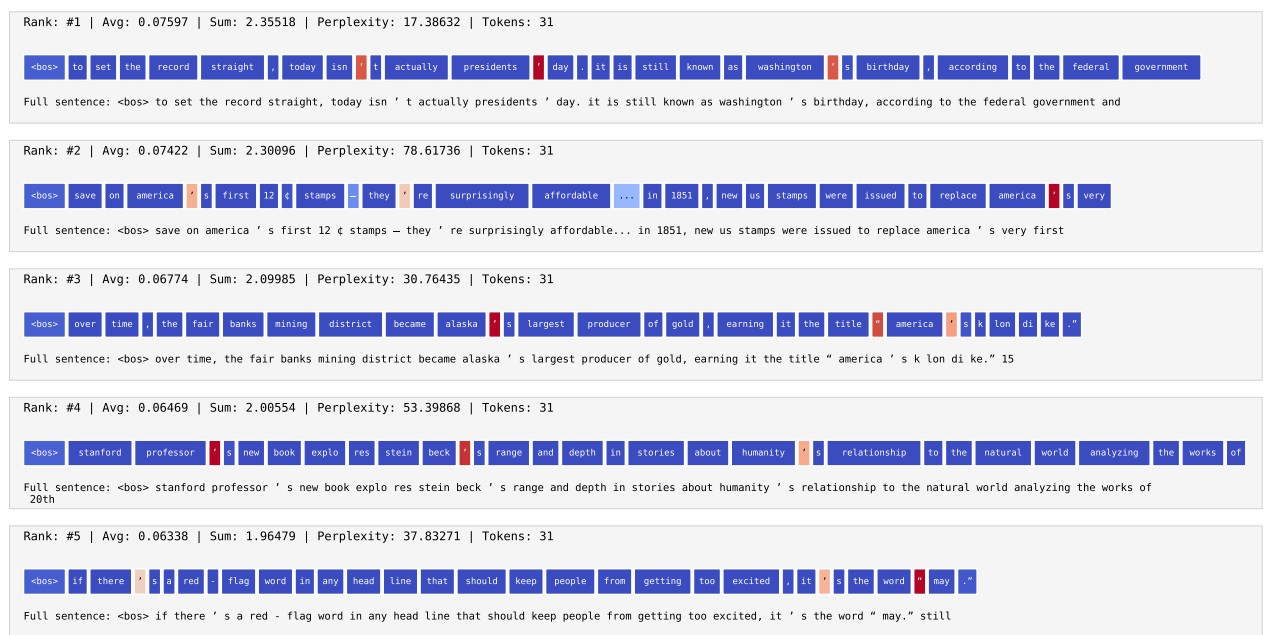

*Figure 5.* Visualization for the control prototype **L9 P2**

communities."

- *"Did you know that mentorship networks are being created to support women and girls in pursuing careers in engineering and mathematics?"*

From this corpus, we defined our test cases. Each case consists of a context (the sentence preceding a target word) and a completion token (the target word itself). For this study, we focused on the probability of the target completions 'women' and 'girls'.

**Results:** After establishing a baseline probability for each test case using the unmodified model, we create a copy of the model for each intervention. The intervention method used is Disruption, where the parameter vector of the target prototype (L9 P7, L9 P18, or L9 P2) is re-initialized with random noise, scaled according to the model's original initialization scheme. This procedure erases the prototype's learned knowledge while preserving the overall model architecture. We then measure the post-intervention probability of the completion token.

The results of our intervention experiments are summarized in Table 7. To focus the analysis on contexts where the target word is considered a plausible completion by the model, we excluded test cases where the baseline probability of the target completion was below 1%.

For prototype re-initialization, our results reveal a clear causal link between prototype *L9 P7* and the model's representation of female-coded concepts. Disrupting this 'female' prototype significantly *decreased* the probability of target words like 'women' ($-16.60\%$) and 'girls' ($-10.67\%$), particularly in less constrained contexts. This effect, however, diminished in test cases where the baseline probability was already very high (e.g., $> 60\%$), suggesting that highly predictable completions are more robust and less reliant on any single prototype. The specificity of this function was confirmed by a control experiment where disrupting an unrelated prototype, *L9 P2*, yielded only negligible changes, proving our findings are not artifacts of random model perturbations. Furthermore, the interventions uncovered a more sophisticated dynamic: disrupting the 'male' prototype, *L9 P18*, consistently *increased* the probability of female-coded words. This suggests an inhibitory or competitive relationship, where the model refines its predictions by balancing between opposing semantic concepts. Taken together, these results demonstrate that the model utilizes specific, functionally distinct, and interacting prototypes to represent and manipulate complex concepts like gender.

For the gate-specific interventions, our results demonstrate that zeroing out the Write or Read gates provides a more

*Table 7.* Comprehensive intervention results. We report the relative change in target probability (%) and sequence perplexity (%) formatted as $\Delta Prob$ ($\Delta PPL$) under three conditions: *Rnd* (Random Re-initialization), *Wr* (Write Gate Mask), and *Rd* (Read Gate Mask). *L9 P7* is the target 'female' prototype; L9 P18 is the 'male' prototype, and L9 P2 serves as a control.

| Context (Truncated) | Base Pr / PPL | L9 P7 ('female') | | | L9 P18 ('male') | | | L9 P2 (Control) | | |
| --- | --- | --- | --- | --- | --- | --- | --- | --- | --- | --- |
| | | *Rnd* | *Wr* | *Rd* | *Rnd* | *Wr* | *Rd* | *Rnd* | *Wr* | *Rd* |
| **Target: 'women'** | | | | | | | | | | |
| ...inequalities of... | 3.21 / 37.58 | **-17.80 (+0.39)** | **-16.60 (+0.29)** | +7.57 (-0.46) | +11.50 (-0.33) | +16.95 (-0.58) | +12.71 (-0.34) | +0.74 (+0.09) | +2.30 (+0.08) | -0.61 (+0.26) |
| ...empowering... | 4.24 / 33.68 | -3.00 (-0.23) | +1.43 (-0.52) | -3.06 (+0.24) | -0.13 (+0.18) | +7.09 (-0.22) | -0.18 (+0.17) | -0.17 (+0.01) | +0.35 (-0.07) | -0.33 (-0.01) |
| ...protect rights of... | 13.54 / 29.82 | +1.37 (-0.16) | -2.29 (+0.14) | +1.56 (-0.09) | +1.43 (-0.06) | -1.03 (+0.14) | +1.64 (-0.08) | +0.09 (+0.03) | -0.22 (-0.00) | +0.38 (+0.00) |
| ...equal opps for... | 10.14 / 75.80 | -0.67 (-0.14) | +3.45 (-0.41) | -0.76 (+0.12) | -0.31 (+0.19) | +2.20 (+0.14) | -0.31 (+0.21) | -0.75 (+0.07) | +0.49 (-0.00) | -1.08 (+0.08) |
| ...climate affects... | 11.87 / 88.10 | +1.81 (-0.31) | -1.65 (+0.09) | +1.86 (-0.12) | +0.12 (-0.13) | -2.61 (+0.21) | +0.22 (-0.11) | +0.34 (-0.15) | -0.32 (+0.14) | +0.58 (-0.18) |
| **Target: 'girls'** | | | | | | | | | | |
| ...inequalities of... | 2.80 / 37.58 | **-10.62 (+0.39)** | **-10.67 (+0.29)** | +5.49 (-0.46) | +0.50 (-0.34) | -1.57 (-0.58) | +0.19 (-0.34) | +0.03 (+0.09) | -0.46 (+0.08) | +0.03 (+0.26) |
| ...empowering... | 68.55 / 33.68 | +0.11 (-0.22) | +0.39 (-0.52) | +0.23 (+0.24) | +0.28 (+0.14) | +0.79 (-0.22) | +0.39 (+0.17) | -0.28 (+0.01) | +0.02 (-0.07) | -0.26 (-0.01) |
| ...protect rights of... | 78.63 / 29.82 | -0.45 (-0.16) | -0.41 (+0.14) | +0.76 (-0.09) | +0.64 (-0.06) | +0.70 (+0.14) | +0.64 (-0.08) | -0.04 (+0.04) | -0.03 (-0.00) | -0.13 (+0.00) |
| ...equal opps for... | 60.49 / 75.80 | -0.17 (-0.13) | +0.01 (-0.41) | +0.58 (+0.12) | +0.56 (+0.18) | +0.84 (+0.14) | +0.65 (+0.21) | -0.19 (+0.07) | +0.02 (-0.00) | -0.27 (+0.08) |
| ...improve services... | 64.33 / 64.39 | -1.56 (-0.45) | -1.43 (-0.43) | +0.35 (+0.11) | +0.62 (+0.24) | +0.94 (+0.00) | +0.72 (+0.27) | -0.15 (-0.07) | +0.05 (+0.12) | -0.19 (-0.09) |
| ...climate affects... | 68.66 / 88.10 | -1.01 (-0.32) | -0.95 (+0.09) | +0.72 (-0.12) | +1.39 (-0.13) | +1.50 (+0.21) | +1.52 (-0.11) | -0.10 (-0.15) | -0.05 (+0.14) | -0.21 (-0.18) |
| ...support women in... | 38.32 / 45.32 | -3.89 (-0.14) | -3.51 (-0.45) | +1.68 (+0.12) | +2.39 (+0.01) | +4.15 (-0.32) | +2.64 (+0.04) | -0.55 (+0.06) | -0.11 (-0.11) | -1.10 (+0.10) |

rigorous measure of causal influence. By acting as a deterministic ablation rather than a stochastic disruption, Write Gate masking revealed a significantly sharper functional contrast between the opposing gender concepts. As shown in Table 7, the divergence between the inhibitory effect of the 'male' prototype (L9 P18) and the causal necessity of the 'female' prototype (L9 P7) was markedly amplified under the masking condition. Specifically, in contexts such as *"...inequalities of..."*, the gap between the probability surge caused by masking the 'male' prototype (+16.95%) and the drop caused by masking the 'female' prototype (-16.60%) was substantially wider ($\Delta \approx 33.6\%$) compared to the spread observed under random disruption ($\Delta \approx 29.3\%$). These findings suggest that strict channel ablation effectively isolates the distinct semantic mechanisms (whether competitive or constructive) that prototypes engage in, with the Write Gate often serving as the primary causal bottleneck for concept storage.

To further assess the specificity of these interventions, we report the sequence perplexity delta ($\Delta PPL$) alongside probability changes in Table 7. We observe that even when a prototype intervention causes a significant drop in target probability (e.g., $-17.80\%$ for 'women'), the impact on global sequence coherence is negligible, with PPL increasing by only +0.39%. This indicates that ProtoT's prototypes are not only functionally relevant but also highly disentangled. Unlike dense models where editing a concept often leads to "whack-a-mole" effects or catastrophic fluency loss (Yang et al., 2024b; Vu & Nguyen, 2025), ProtoT allows for surgical interventions with minimal collateral damage to the model's overall predictive stability.

### A.2.2. ROBUSTNESS OF CONCEPT-SPECIFIC PROTOTYPES EMERGENCE ACROSS RANDOM SEEDS

To ensure that the localization of semantic concepts is a robust property of the architecture rather than an artifact of a specific initialization, we extended our analysis by training two additional models with different random seeds. We repeated the visualization process described in Sec. 5.1 for these new runs to observe if similar semantic clusters emerged. We focus on the concept of *'COVID-19'* as a representative case study. In all three models (the original and two replicates), we successfully identified a distinct prototype that was maximally activated by terms related to the pandemic.

To validate the functional consistency of these re-emerged prototypes, we conducted intervention experiments targeting the prediction of the token *'COVID'* in relevant contexts. For each model, we disrupted the identified COVID-specific prototype via random re-initialization, write gate masking and read gate masking. We followed the same workflow as in Sec. A.2.1. The test corpus for the COVID-specific prototype consisted of the following sentences:

- *"covid - 19 lambda variant lambda variant cases of covid - 19 are emerging in the us. while nowhere near"*

- *"The World Health Organization declared the outbreak of COVID-19 a pandemic in March 2020."*

- *"Researchers identified the Alpha, Beta, Gamma, and Delta strains as variants of concern for COVID-19."*

- *"The Pfizer-BioNTech and Moderna vaccines use mRNA technology to protect against the COVID-19 virus."*

- *"Anosmia, the sudden loss of smell and taste, was identified as a specific symptom of COVID-19 infection."*

- *"To curb the spread, the government mandated a 14-day quarantine for anyone testing positive for COVID-19."*

- *"Hospitals faced a critical shortage of ventilators during the initial surge of severe COVID-19 cases."*

- *"The FDA granted emergency use authorization for Paxlovid, an oral antiviral pill for treating COVID-19."*

- *"Scientists continue to debate the zoonotic origins of COVID-19 and its potential transmission from bats."*

- *"Despite strict border controls, the Omicron variant of COVID-19 managed to spread rapidly across the globe."*

- *"Long-haulers are patients who suffer from debilitating symptoms months after recovering from acute COVID-19."*

- *"Public health officials urged the population to wear N95 masks to prevent the airborne transmission of COVID-19."*

- *"The CDC updated its guidelines regarding the isolation period for asymptomatic cases of COVID-19."*

- *"Herd immunity against COVID-19 became difficult to achieve due to the emergence of new escape variants."*

- *"Schools implemented social distancing and improved ventilation to reduce the risk of COVID-19 transmission in classrooms."*

- *"The economic fallout from the COVID-19 pandemic led to supply chain disruptions and rising inflation."*

- *"A negative PCR test result for COVID-19 was required for all passengers boarding international flights."*

- *"Studies suggest that previous infection provides some level of natural immunity against reinfection with COVID-19."*

- *"The global death toll attributed to COVID-19 has highlighted the vulnerabilities in healthcare systems worldwide."*

- *"Contact tracing apps were deployed to alert individuals who had been exposed to a confirmed case of COVID-19."*

- *"Rehabilitation programs are being established to help patients recover from the respiratory damage caused by severe COVID-19."*

*Table 8. Cross-seed consistency of the 'COVID' prototype with perplexity.* We compare the impact of interventions across three different model initializations. Cells show relative change in target probability (%) and perplexity (%) formatted as $\Delta Prob$ ($\Delta PPL$). Cells with '–' indicate that the baseline probability was below the 1% threshold for reliability.

| Context (Truncated) | Original Model (L1 P14) | | | | Seed 124 (L7 P29) | | | | Seed 325 (L6 P31) | | | |
|---|---|---|---|---|---|---|---|---|---|---|---|---|
| | Base Pr / PPL | Rnd | Wr | Rd | Base Pr / PPL | Rnd | Wr | Rd | Base Pr / PPL | Rnd | Wr | Rd |
| covid – 19 lambda variant... | 12.36 / 40.68 | +0.32 (+0.09) | +0.52 (-0.30) | -0.02 (+0.26) | 48.67 / 28.46 | -8.53 (+1.35) | -8.36 (+1.23) | +0.08 (-0.00) | 27.37 / 41.70 | +4.51 (+0.56) | +6.31 (-0.13) | -5.67 (-0.81) |
| The World Health Organization... | 6.42 / 12.89 | -1.60 (+0.13) | -1.72 (+0.08) | +0.24 (-0.03) | 16.17 / 10.98 | -0.41 (+0.32) | +0.05 (+0.20) | -0.09 (+0.00) | 9.65 / 13.97 | +12.62 (+1.06) | +12.96 (+1.71) | +13.08 (+0.44) |
| The Pfizer–BioNTech and Moderna... | 0.62 / 24.06 | – | – | – | 0.95 / 24.46 | – | – | – | 1.40 / 27.51 | +0.33 (+0.46) | -0.37 (+0.94) | +0.13 (+0.47) |
| To curb the spread,... | 10.44 / 28.96 | -0.23 (+0.14) | -0.59 (+0.11) | +0.02 (-0.02) | 13.30 / 30.12 | -0.90 (+0.19) | -0.95 (+0.21) | +0.43 (-0.02) | 17.33 / 31.78 | -0.62 (-0.25) | +0.31 (-0.45) | -0.44 (+0.10) |
| Hospitals faced a critical... | 0.40 / 42.40 | – | – | – | 1.35 / 28.02 | +1.69 (+0.11) | +2.02 (+0.09) | -0.10 (+0.00) | 2.37 / 33.74 | +23.35 (-1.52) | +27.32 (-0.46) | +22.14 (-2.28) |
| The FDA granted emergency... | 0.70 / 59.13 | – | – | – | 3.44 / 72.11 | +0.84 (-0.44) | +0.83 (-0.35) | -0.11 (-0.03) | 1.05 / 59.34 | +9.31 (+0.47) | +5.94 (+1.02) | -3.81 (-0.28) |
| Despite strict border controls,... | 0.70 / 36.71 | – | – | – | 2.22 / 30.61 | -3.69 (-0.20) | -3.58 (-0.22) | +0.00 (+0.00) | 2.41 / 28.36 | +3.00 (-0.65) | +2.57 (-1.04) | +2.19 (-0.27) |
| Public health officials urged... | 6.01 / 36.44 | +0.25 (+0.02) | -0.18 (+0.05) | -0.06 (+0.00) | 3.32 / 31.63 | +0.48 (+0.28) | +0.46 (+0.24) | -0.03 (-0.05) | 12.72 / 30.33 | -2.70 (+0.50) | -8.10 (+0.02) | -2.90 (+0.69) |
| The CDC updated its... | 2.46 / 48.34 | -0.30 (-0.03) | -0.10 (-0.06) | +0.11 (+0.02) | 3.42 / 40.36 | -0.89 (+0.07) | -0.54 (+0.07) | +0.08 (-0.01) | 3.47 / 40.32 | +3.43 (-0.21) | -10.85 (+0.27) | +5.67 (-0.54) |
| Herd immunity against COVID... | 7.87 / 92.05 | +0.23 (-0.06) | +0.36 (+0.63) | +0.00 (-0.11) | 5.38 / 100.81 | -0.05 (-0.18) | +0.00 (-0.09) | +0.00 (+0.01) | 6.49 / 108.74 | +1.47 (+3.46) | -1.81 (+5.58) | +1.63 (+4.90) |
| Schools implemented social... | 1.79 / 30.28 | -0.37 (+0.10) | -0.36 (+0.19) | +0.13 (-0.12) | 2.31 / 24.76 | +0.47 (+0.52) | +0.87 (+0.57) | -0.01 (-0.01) | 3.98 / 29.50 | +1.36 (+0.03) | -0.91 (-0.30) | +1.33 (-0.08) |
| The economic fallout from... | 2.38 / 28.56 | +1.95 (+0.07) | +2.26 (+0.07) | -0.19 (+0.01) | 1.99 / 21.67 | +0.21 (+0.27) | -0.02 (+0.23) | -0.01 (-0.01) | 1.52 / 24.02 | -4.79 (-1.25) | +3.13 (-2.56) | -5.13 (-1.79) |
| Studies suggest that previous... | 1.32 / 76.24 | +0.58 (+0.12) | +0.92 (+0.10) | -0.05 (-0.04) | 1.02 / 66.88 | -0.07 (+0.06) | +0.22 (+0.05) | +0.06 (-0.01) | 2.47 / 80.82 | +0.03 (+2.99) | -2.92 (+3.13) | +0.35 (+3.35) |
| The global death toll... | 2.68 / 44.60 | +0.42 (-0.03) | +0.28 (-0.04) | +0.00 (-0.01) | 2.85 / 49.93 | +0.10 (-0.69) | -0.04 (-0.55) | +0.01 (+0.01) | 1.59 / 43.93 | -6.53 (+0.47) | -13.42 (+0.34) | -6.78 (+0.53) |
| Contact tracing apps were... | 2.34 / 37.09 | +0.32 (-0.08) | +0.41 (-0.11) | -0.13 (+0.02) | 4.00 / 43.90 | -0.14 (+0.17) | +0.19 (+0.19) | +0.07 (-0.00) | 5.11 / 39.25 | +0.25 (+1.61) | -0.26 (-0.19) | +1.58 (+1.48) |
| Rehabilitation programs are... | 0.22 / 48.34 | – | – | – | 0.93 / 34.98 | – | – | – | 2.89 / 33.86 | +3.93 (+1.45) | -21.97 (+0.69) | +3.98 (+1.08) |

**Results:** Our analysis across different random seeds shows that the ProtoT architecture consistently localizes concepts within specific semantic regions, demonstrating strong robustness. However, the specific functional mechanisms used by these prototypes can differ significantly between model initializations. We acknowledge that the intervention effects for the COVID-19 prototype appear more variable than those observed in the gender experiments. This difference comes in part from the way the concepts are structured. Gender is modeled as a clear binary contrast (Male vs. Female), which allows relative comparisons, while COVID-19 is a single concept – it depends only on changes from a baseline probability and has no direct opposite reference. Despite this lack of contrastive referencing, the most significant finding is the robust emergence of the concept itself: across all three random seeds, the model consistently allocated a dedicated prototype slot to encode pandemic- or disease-related knowledge without explicit supervision.

A.2.3. EXTENDED INTERVENTION ANALYSIS ON DIVERSE CONCEPTS

To demonstrate that the functional localization of semantic concepts is a general property of the ProtoT architecture and not limited to the social (Gender) or event-specific (COVID-19) cases discussed in the above, we extended our discovery pipeline to additional semantic domains. Here, we present intervention results for geographic entity (*"e.g. New Zealand"*) and an abstract state (*"e.g. Mental Health"*).

The test corpus for the 'New Zealand' prototype consisted of the following sentences:

- *"The Dutch explorer Abel Tasman named New Zealand as Nova Zeelandia after the Dutch province of Zeeland"*

- *"The Treaty of Waitangi signed in 1840 was instrumental in establishing British sovereignty over New Zealand"*

- *"Regular Quaker meetings began in Nelson in 1842 and later spread across New Zealand"*

- *"The first Quaker to visit Aotearoa / New Zealand was Sydney Parkinson"*

The test corpus for the 'Mental Health' prototype consisted of the following sentences:

- *"Meditation and mindfulness practices are beneficial for maintaining mental clarity."*

- *"Regular exercise can improve mental well-being and reduce symptoms of anxiety."*

- *"She sought professional help to manage her mental stress during the exam period."*

- *"Many people face mental health challenges but do not seek support due to stigma."*

We identified a prototype in L5 P9 that maximally activated for contexts related to the country including *New Zealand*. To validate its causal role, we constructed a test set containing historical and geographical facts. We then measured the impact of masking the prototype to see the change on the prediction of the target token *'zealand'*.

The results (Table 9) show striking causal efficacy. For instance, in the context of *"...abel tasman named new [zealand]"*, masking the write gate caused a massive probability drop of 21.54%. Similarly, references to the *Treaty of Waitangi* saw a 12.52% drop. This confirms that this specific slot (L5 P9) is critical for storing and retrieving knowledge specific to this geographic entity.

*Table 9. Intervention results for the 'New Zealand' prototype (L5 P9).* Cells show relative change in target probability (%) and perplexity (%) formatted as $\Delta Prob$ ($\Delta PPL$).

| Context (Truncated) | Base Pr / PPL | L5 P9 ('New Zealand') | | |
| --- | --- | --- | --- | --- |
| | | *Rnd* | *Wr* | *Rd* |
| the dutch explorer abel tasman... | 35.67 / 199.89 | -20.67 (+3.45) | -21.54 (+3.47) | -21.03 (+3.55) |
| the treaty of waitangi signed... | 91.86 / 54.97 | -12.41 (+2.31) | -12.52 (+2.39) | -12.93 (+2.56) |
| regular quaker meetings began... | 33.42 / 377.04 | -1.11 (+0.76) | -2.27 (+0.95) | -1.24 (-0.38) |
| the first quaker to visit... | 43.05 / 245.31 | -3.49 (-0.35) | -3.50 (-0.33) | -3.85 (-0.04) |

Moving beyond concrete entities, we investigated whether abstract concepts are similarly localized. We identified a prototype in L6 P9 responsive to *Mental Health* and communication. We tested this using sentences involving psychological states and well-being, targeting the token *'mental'*.

As shown in Table 10, while the baseline probabilities for this abstract adjective are generally lower than for proper nouns, intervention still yields consistent causal effects, further supporting the functional diversity discussed in the main text.

Crucially, the perplexity trends observed in the cross-seed robustness trials (Table 8) and the domain-extended interventions (Table 9 and Table 10) strongly replicate the insights derived from our gender-based experiments. Across all the experiments, the overall change in perplexity stays remarkably small. This consistency across very different domains provides clear quantitative evidence that ProtoT's interventions remain highly selective and minimally disruptive by design, rather than being artifacts of a particular task. In turn, this highlights ProtoT's potential as a clean and reliable approach for behavior modification.

*Table 10. Intervention results for the 'mental' prototype (L6 P9).* Cells show relative change in target probability (%) and perplexity (%) formatted as $\Delta Prob$ ($\Delta PPL$).

| | | L6 P9 ('mental') | | |
|---|---|---|---|---|
| Context (Truncated) | Base Pr / PPL | Rnd | Wr | Rd |
| Meditation and mindfulness... | 5.04 / 78.83 | -1.56 (-0.66) | -2.20 (-0.92) | -2.31 (-1.12) |
| Regular exercise can improve... | 2.06 / 45.84 | -0.62 (-3.13) | -0.44 (-3.01) | -0.69 (+0.36) |
| She sought professional help... | 1.35 / 268.86 | -0.17 (-0.90) | -0.18 (-0.85) | -0.16 (-0.14) |
| Many people face mental... | 1.32 / 77.17 | -0.01 (-0.05) | -0.09 (-0.04) | -0.01 (+0.04) |

*Table 11.* GLUE dev downstream fine-tuning results (all metrics reported as percentages). For COLA we report Matthews correlation; for SST-2 accuracy; for MRPC F1; for STS-B Pearson correlation; for RTE, WNLI, QNLI, MNLI and MNLI-MM accuracy; for QQP F1. Results are averaged over 3 seeds. Best results are in bold.

| Model | COLA | SST-2 | MRPC | STS-B | RTE | WNLI | QQP | QNLI | MNLI | MNLI-MM |
|---|---|---|---|---|---|---|---|---|---|---|
| LLaMA | **36.4** | **91.2** | **84.9** | **85.8** | **60.1** | **56.3** | **85.9** | **86.4** | **80.5** | **79.7** |
| Mamba | 30.6 | 89.1 | 82.3 | 79.4 | 55.8 | **56.3** | 82.7 | 82.2 | 75.1 | 74.9 |
| DeltaNet | 8.9 | 85.5 | 81.4 | 75.7 | 56.1 | **56.3** | 81.2 | 80.1 | 71.6 | 72.5 |
| ProtoT | 29.3 | 89.8 | 81.7 | 73.3 | 54.5 | 54.9 | 83.6 | 82.6 | 75.4 | 76.0 |

## A.3. Downstream (Details)

We provide the training protocol and hyperparameter configuration used for the GLUE downstream experiments, covering datasets and splits, pre-processing, optimization, early-stopping/selection on dev, and the hyperparameter sweep and choice rules, and we additionally report the corresponding GLUE dev-set results for completeness.

**Training protocol:** We evaluate four language model architectures: ProtoT, LLaMA, Mamba, and DeltaNet, on the GLUE benchmark under a unified experimental protocol to ensure fair comparison. Unless stated otherwise, all details follow the Experimental Setup 4. Inputs are formed as single-sentence or sentence-pair prompts according to the task, with a maximum sequence length of 512. To avoid leakage, we fine-tune on the official training split, select hyperparameters and checkpoints on the official development split using early stopping, and export test predictions in the official TSV format for submission to the GLUE server. We follow the official GLUE metrics: accuracy for SST-2, QNLI, MNLI, QQP, RTE, and WNLI (or the primary metric reported by the official script), the accuracy and F1 pair for MRPC and QQP, Matthews correlation for CoLA, and Pearson and Spearman correlations for STS-B.

Optimization and regularization are aligned across models. We use the AdamW optimizer together with a linear learning-rate schedule with warmup. We apply selective weight decay consistent with pre-training: decay is applied to affine weights that benefit from it, while embeddings, normalization layers, and biases receive no decay. For GLUE downstream fine-tuning, we use a batch size of 16 for all models. Compared to pre-training, the GLUE datasets are much smaller, so we prefer a moderately small batch size that provides more stochasticity in the updates and typically leads to better generalization in low-data regimes. Fine-tuning runs for up to 3 epochs with early stopping on dev, and the dev-best checkpoint is used to generate test predictions. Unless otherwise specified, a fixed random seed is used across tasks and models to support reproducibility.

**Hyperparameter selection:** Because architectures differ in optimization sensitivity, we conduct per-model hyperparameter selection. For each model we run small grid searches on two representative tasks, SST-2 (medium-scale binary classification) and MNLI (large-scale multi-class classification). We sweep learning rates over a grid that includes 2.5e-5, 3.5e-5, 5.5e-5, 1e-4, 2e-4, 3e-4, 4e-4, 5e-4, 7e-4, 8.5e-4, 1e-3, and we sweep warmup ratios over 6% and 10%. The best learning rate and warmup found per model on these representative tasks are then fixed for that model across the remaining GLUE tasks, where "best" is defined as the (learning-rate, warmup) configuration that achieves the highest average dev performance over SST-2 and MNLI. All other training details, such as batch size, maximum length, optimizer settings, and early-stopping criterion, remain identical across models.

The final per-model settings in our environment are as follows. PrototypeAttn uses a learning rate of 3.5e-5 with 6% warmup. LLaMA uses a learning rate of 5.5e-5 with 10% warmup. Mamba uses a learning rate of 1e-4 with 10% warmup. DeltaNet uses a learning rate of 7e-4 with 10% warmup.

**GLUE dev downstream fine-tuning results:** For completeness, we also report the GLUE dev set results in Table 11, using the same evaluation metrics as in the main text (Matthews correlation for CoLA, accuracy for SST-2/RTE/WNLI/QNLI/MNLI/MNLI-MM, F1 for MRPC and QQP, and Pearson correlation for STS-B). These dev numbers were used during model development and are largely consistent with the test-set trends in the results in Section 5, Table 3: LLaMA achieves the strongest overall performance, while ProtoT remains competitive with dense baselines and shows robust behavior across multiple tasks.

### A.4. Robustness (Details)

This section details the perturbation set, construction pipeline, and slice-level statistics for the black-box robustness experiments.

#### A.4.1. PERTURBATION DATASET CONSTRUCTION

We construct a dedicated perturbation benchmark with seven categories of *meaning-preserving* surface noise, 500 pairs per category (3,500 total). Source sentences are sampled from three public corpora under simple length and formatting constraints: WikiText-2 (Merity et al., 2017), DailyDialog (Li et al., 2017), and AG News (Zhang et al., 2015).

The final slices are:

- **Synonyms:** Replacements derived from WordNet (Miller, 1995). We select candidate lemmas that differ from the original token, avoid multi-word expressions, and have similar length. The generator enforces a mix of 1/2/3 substitutions per sentence. This slice is further filtered using Sentence-BERT (all-MiniLM-L6-v2; (Reimers & Gurevych, 2019; Wang et al., 2020)) and lexical heuristics (see below).

- **Typos:** Single-character keyboard noise applied to one token (internal character substitution), ensuring short, localized corruption.

- **Spelling variants:** American $\leftrightarrow$ British spelling changes (e.g., *color $\rightarrow$ colour*, *organize $\rightarrow$ organise*) using a fixed mapping and a curated example pool.

- **Morphological variants:** Simple inflection changes (e.g., singular $\rightarrow$ plural) using rule-based morphology patterns plus a curated pool.

- **Contractions/Expansions:** Deterministic mappings between standard and contracted forms (e.g., *do not $\rightarrow$ don't*, *it is $\rightarrow$ it's*).

- **Punctuation/Casing:** Insertion or removal of punctuation (e.g., sentence-final periods or comma adjustments), while keeping word order intact.

- **Abbreviations/Short forms:** Systematic long-form $\rightarrow$ abbreviation mappings (e.g., *Doctor $\rightarrow$ Dr.*, *United States $\rightarrow$ U.S.*).

After generating the dataset, a separate cleaning and rebalancing script was run:

1. filter by Sentence-BERT similarity ($\geq 0.75$),

2. apply lexical checks (rare tokens, casing),

3. backfill missing items from curated fallback pools, and

4. sample exactly 500 pairs per slice.

#### A.4.2. VARIANCE STATISTICS

To characterize the perturbation strength of each slice, we compute: (i) cosine similarity of all-MiniLM-L6-v2 embeddings between original and perturbed sentences, (ii) character-level Levenshtein distance (Levenshtein, 1966).

Table 13 reports the per-slice averages. As expected, some categories (e.g., typos) introduce very small character edits but can cause non-trivial distributional shifts, while others (e.g., synonyms, abbreviations) involve larger form changes yet maintain high semantic similarity.

*Table 12.* Example sentence pairs from the perturbation benchmark.

| Category | Original | Perturbed |
|---|---|---|
| Abbreviation | *Doctor Smith arrived.* | *Dr. Smith arrived.* |
| Contraction | *I cannot go.* | *I can't go.* |
| Synonym | *He was happy.* | *He was glad.* |
| Spelling | *I like this color.* | *I like this colour.* |

*Table 13.* Variance statistics for the perturbation benchmark (3,500 pairs total).

| | Synonym | Typo | Spelling | Morphology | Contraction | Punctuation | Abbreviation |
|---|---|---|---|---|---|---|---|
| Avg. Similarity | 0.828 | 0.775 | 0.956 | 0.881 | 0.895 | 0.983 | 0.894 |
| Avg. Edit Distance | 5.89 | 1.03 | 1.20 | 1.00 | 2.54 | 1.09 | 7.23 |

### A.5. Throughput Benchmarks (ProtoT, Mamba, LLaMA, DeltaNet)

We evaluate under identical conditions: same data pipeline, optimizer, precision (BF16), sequence length 256, and batch sizes 32 and 128. FLOP counts are per-sample (forward+backward) where obtainable. *Observations*: Table 14 summarizes training throughput at batch sizes 32 and 128 for matched-depth/width models. LLaMA attains the highest throughput overall (**55.1 and 23.6** it/s). ProtoT sustains **25.2 and 7.6** it/s and is ~2.1–2.4× faster than Mamba (11.9 and 3.2 it/s) at the same backbone. The FLA-based DeltaNet baseline, evaluated without fused delta kernels and with `torch.compile` disabled, reaches 3.5 and 1.8 it/s (batch 32/128).

#### A.5.1. LONG-CONTEXT THROUGHPUT

**Throughput Evaluation Methodology**  To evaluate the computational efficiency of the models at varying context lengths, we measured the processing throughput on a single NVIDIA A100 80GB GPU. The benchmark measured the number of forward pass iterations per second (it/s) for a batch size of 1 across context lengths ranging from 2,048 to 131,072 tokens.

For a fair comparison, all models were run in standard PyTorch eager mode without `torch.compile` optimization. This ensures that the results reflect the raw architectural performance characteristics rather than compiler-specific optimizations which may vary in maturity across different architectures.

**Throughput Results**  Table 15 shows the long-context inference throughput results. LLaMA achieves the highest throughput at short context lengths. However, ProtoT scales better as context increases, surpassing LLaMA at 32k tokens and above. DeltaNet maintains the highest throughput at long context lengths.

### A.6. Ablations

#### A.6.1. LAYER-0 ROUTING ABLATIONS

In Table 16, we ablate the three mitigations that stabilize the layer-0 router: (i) sharing the write/read routing distribution, (ii) sharpening the initial temperature ($\tau_0 = 3.0$), and (iii) adding a $k = 5$ depth-wise convolution to the write-value path of layers 0–1. Each configuration fine-tunes a 6-layer ProtoT on the FineWeb 18k/4k split (sequence length 256, seed 0) for three epochs, using the same optimizer, tokenizer, and learning rate as the main experiments. We report best validation perplexity alongside routing diagnostics logged on the dev set.

The convolution contributes most to stability: removing it roughly doubles the router concentration (top-1 rises from 0.079 to 0.177), increases hub inequality, and halves the layer-0 ReZero gate, ultimately worsening perplexity by +12.4 points. Shared routing and the sharpened $\tau_0$ have smaller individual effects on perplexity, but together they keep hub utilisation uniform (gini 0.034) while allowing the gate to stay near its baseline value. Disabling every mitigation reproduces the original alpha-collapse, dropping $\bar{\alpha}_0$ to 0.261 and letting a single hub monopolise 37% of the mass.

*Table 14.* Training throughput (it/s; higher is better) and elapsed time (s; lower is better) for matched-depth/width models at seq. len. 256 (BF16). FLOPs are reported in units of $\times 10^5$ (forward+backward). *When compilation was unavailable, values reflect the fastest steady-state runs without compilation.*

| Model | Batch | it/s | Elapsed (s) | FLOPs/sample ($\times 10^5$) | Total FLOPs ($\times 10^5$) | Params |
|---|---|---|---|---|---|---|
| ProtoT | 32 | 25.2 | 34.57 | 41,583.0 | 1,330,657.0 | 12,205,266 |
| ProtoT | 128 | 7.6 | 31.32 | 41,583.0 | 5,322,625.2 | 12,205,266 |
| Mamba | 32 | 11.9 | 58.17 | **34,734.9** | **1,111,517.4** | **6,724,352** |
| Mamba | 128 | 3.2 | 54.26 | **34,734.9** | 4,446,069.4 | **6,724,352** |
| DeltaNet | 32 | 3.5 | 222.88 | — | — | 12,963,456 |
| DeltaNet | 128 | 1.8 | 182.06 | — | — | 12,963,456 |
| LLaMA | 32 | **55.1** | 26.16 | 49,341.5 | 1,578,929.0 | 12,938,496 |
| LLaMA | 128 | 23.6 | **22.30** | 49,341.5 | 6,315,714.3 | 12,938,496 |

*Table 15.* Long-context inference throughput (iterations per second; higher is better). Measured on a single NVIDIA H100 80GB GPU, batch size 1, PyTorch eager mode without `torch.compile`.

| Context | LLaMA | ProtoT | Mamba | DeltaNet |
|---|---|---|---|---|
| 2,048 | 100.47 | 38.74 | 35.50 | 43.54 |
| 4,096 | 47.90 | 20.61 | 19.04 | 42.35 |
| 8,192 | 21.72 | 10.82 | 9.89 | 44.89 |
| 16,384 | 8.08 | 5.49 | 5.05 | 40.20 |
| 32,768 | 2.61 | 2.78 | 2.55 | 27.78 |
| 65,536 | 0.74 | 1.40 | 1.28 | 17.34 |
| 131,072 | 0.20 | 0.65 | 0.57 | 9.41 |

*Table 16.* Layer-0 routing ablations on FineWeb. Metrics come from the final validation epoch (`val_router_stats.csv`) and the best dev perplexity tracked during training. Lower perplexity, Gini, and top-1 probability imply healthier routing; higher $\bar{\alpha}_0$ indicates that the ReZero gate remains active. Best values are in bold.

| Variant | Shared $L_0$ | $\tau_0$ init | Write conv | Best val ppl $\downarrow$ | $\bar{\alpha}_0 \uparrow$ | Gini $\downarrow$ | top-1 $\downarrow$ |
|---|---|---|---|---|---|---|---|
| All mitigations (baseline) | On | 3.0 | $k = 5$ | **133.3** | **0.672** | **0.034** | **0.079** |
| No shared routing | Off | 3.0 | $k = 5$ | 133.4 | 0.658 | 0.064 | 0.082 |
| $\tau$ reset to 1.0 | On | 1.0 | $k = 5$ | 133.6 | 0.653 | 0.035 | 0.088 |
| No write conv | On | 3.0 | Off | 145.7 | 0.354 | 0.097 | 0.177 |
| All mitigations off | Off | 1.0 | Off | 149.9 | 0.261 | 0.243 | 0.373 |

**Interpretation.** Shared write/read routing and the sharper initial temperature primarily act as regularisers: they prevent the router from collapsing mass onto a few hubs without hurting sample efficiency. The depth-wise convolution, in contrast, provides an expressivity boost that both improves perplexity and raises the effective signal scale entering layer 0; once it is removed the router cannot maintain broad support and the ReZero gate decays. The combination of all three mitigations therefore offers a balanced trade-off between stability and performance.

### A.6.2. VALIDATING THE HYPERPARAMETER CHOICES

These experiments motivate the choice of kernel size (5) for the local convolution, the alpha-gate initialization (1.0), the number of prototypes (32), the use of mass normalization, low-rank projection at the value stream, and dropout. In these experiments, we use the default model, data, and training configurations, unless otherwise specified. We search over learning rates values (1.0e-3, 2.0e-3, 3.0e-3) for ProtoT, (0.8e-3, 1.6e-3, 3.2e-3) for LLaMA, (1.9e-3, 3.8e-3, 7.6e-3) for Mamba, and (3.4e-3, 6.8e-3, 13.6e-3) for DeltaNet (the middle values of each interval are informed by the best learning rates from the automatic hyperparameter search, Section 4), and average the results over 3 seeds.

**Kernel size of local convolution** The results in Table 17 show that kernel size 5 and 6 are the top 2 values in terms of dev perplexity, with an insignificant difference ($\approx$0.4%, within cross-seed variation) between the two (94.3 vs 93.9), which

confirms our choice of kernel size = 5.

*Table 17.* Kernel size ablation of the local convolution in ProtoT. Reported best dev perplexity (lower is better), averaged over 3 seeds. Best values are in bold.

| Variant | Kernel size | Performance (dev perplexity) ↓ |
|---|---|---|
| ProtoT (k=4) | 4 | 94.5 |
| ProtoT (k=5) | 5 | 94.3 |
| ProtoT (k=6) | 6 | **93.9** |
| ProtoT (k=7) | 7 | 94.6 |

**Alpha-gate initialization:** The results in Table 18 show that 1.0 is the best values for $\alpha$ initialization, which confirms our choice. In particular, $\alpha = 1.0$ performs slightly better than ReZero's $\alpha = 0.0$ (Bachlechner et al., 2021), with 93.8 vs 94.5 perplexity. This is likely because ReZero trains extremely-deep NNs, where it may be beneficial to start from zero contribution from the layers, to avoid noise accumulation early in training.

*Table 18.* Alpha-gate initialization study for ProtoT. Reported best dev perplexity (lower is better), averaged over 3 seeds. Best values are in bold.

| Alpha initialization value | Performance (dev perplexity) ↓ |
|---|---|
| 0.0 | 94.5 |
| 0.5 | 94.9 |
| 0.8 | 94.3 |
| 1.0 | **93.8** |
| 1.2 | 95.4 |

**Mass Normalization and Low-Rank Projection at the Value Stream:** The results in Table 19 show that the mass normalization (used in the default setting) brings ≈9% slowdown, while improving perplexity by ≈3.3% (97.7→94.5). This is a trade-off, where we have chosen the performance gain over the slowdown. On the other hand, the low-rank projection to half the hidden size performs similarly to using the full hidden size in terms of dev perplexity (94.5 vs 95.2, within per-seed variation), while introducing a massive speed-up (≈59% faster), thus justifying our choice.

*Table 19.* Ablation study for ProtoT: mass normalization and low-rank projection. Reported best dev perplexity (lower is better), averaged over 3 seeds. Best values are in bold.

| Setting | Performance (dev perplexity) | Speed after torch.compile() [it/s] |
|---|---|---|
| Default settings | **94.5** | 89 |
| No mass normalization | 97.7 | **98** |
| No low-rank projection | 95.2 | 56 |

**Optimal number of prototypes (R):** The results in Table 20 show that ProtoT's performance gains significantly diminish after R=32 (94.0→93.5 perplexity), whereas speed drops substantially (89→60 it/s). This shows that R=32 is the optimal trade-off between model performance and speed.

*Table 20.* Optimal number of prototypes (R) for ProtoT. Reported best dev perplexity (lower is better), averaged over 3 seeds. Best values are in bold.

| R value | Performance (dev perplexity) | Speed after torch.compile() [it/s] |
|---|---|---|
| 16 | 95.9 | **116** |
| 32 | 94.0 | 89 |
| 64 | **93.5** | 60 |

**Dropout:** The results in Table 21 show that the default dropout configuration we use in Section 4 yields the best perplexity for ProtoT and all baseline models.

*Table 21.* Dropout study for ProtoT, LLaMA, Mamba, and DeltaNet. Reported best dev perplexity (lower is better), averaged over 3 seeds. Best values are in bold.

| Model settings | Performance (dev perplexity) ↓ |
|---|---|
| ProtoT (default: dropout=0.1) | **94.5** |
| ProtoT (no dropout) | 103.8 |
| LLaMA (default: dropout=0.1) | **81.0** |
| LLaMA (no dropout at self-attn) | 83.5 |
| LLaMA (no dropout at all) | 89.1 |
| Mamba (default: dropout=0.1) | **89.8** |
| Mamba (no dropout) | 95.4 |
| DeltaNet (default: dropout=0.1) | **93.3** |
| DeltaNet (no dropout) | 94.5 |

## A.7. Additional Interpretability Metrics

In this appendix, we report results on correlation between half life values and locality of a concept as well as four complementary metrics that characterize how prototype activations of ProtoT and LLaMA attention heads value norms evolve across depth. Each metric captures a different aspect of how the routing distribution changes from early to deeper layers. Let $a_{l,p}(x)$ denote the activation of prototype $p \in \{1, \ldots, P\}$ at layer $l \in \{1, \ldots, L\}$ for input $x$, and let $\mathbf{a}_l(x) = (|a_{l,1}(x)|, \ldots, |a_{l,P}(x)|)$ denote the vector of absolute activations.

**Correlation Between Half-Life and Locality** To assess whether prototype half-life reflects concept locality, we require an operational proxy for locality. Empirically, low-level lexical prototypes (e.g., punctuation, stopwords) exhibit highly repetitive sets of most-activating-tokens, whereas more abstract prototypes show greater token diversity. This aligns with the intuition that local prototypes focus on neighboring tokens and as a result, activate more strongly based on token identity, while longer half-life prototypes aggregate information over wider contexts. Motivated by this observation, we use *token repetition score* as a proxy for locality. we use *token repetition score* as a proxy for locality. We define the repetition score as the proportion of non-unique tokens:

$$\text{repetition\_score} = \frac{|T| - |U|}{|T|} = 1 - \frac{|U|}{|T|} \tag{3}$$

where $|T|$ is the total number of tokens among the most-activating examples and $|U|$ is the number of unique token types. Higher scores indicate more repetitive (local) patterns.

| **Low half-life prototype (L2P6)** | **Higher half-life prototype (L2P10)** |
|---|---|
| HL = 5.04, repetition = 0.73 | HL = 12.26, repetition = 0.25 |
| Function-word clusters | Transformation expressions |
| (e.g., "in the", "of the") | (e.g., "from · to ·") |

```
the , the , ,                          guide for aim deliver their close
in the , the of , ,                    converted to homeless converted an into
a many of the , ,                      adaptation of through applied to the
, or of the most and                   delegation responsibility shifts from ``common''
the a , in , this                      delegation
in the , in other ,                    manage .  bring our home from
                                       uses encourage invite your to build
```

*Figure 6.* Two example prototypes illustrating the relationship between half-life and repetition. The short half-life prototype (left) exhibits tightly localized, highly repetitive lexical patterns, whereas the longer half-life prototype (right) activates on broader transformation expressions.

We quantify the relationship between half-life and repetition using two standard statistical tools: (i) a Spearman rank

correlation between half-life and repetition scores, and (ii) quantile-based group comparisons in which prototypes are divided into half-life quartiles. For the latter, we compare mean repetition scores across quartiles and compute the effect size (Cohen's $d$) between the lowest and highest half-life groups.

| Metric | Value | Interpretation |
|---|---|---|
| Spearman $\rho$ | $-0.2192$ | Negative association |
| $p$-value | $8.3 \times 10^{-10}$ | Very significant |
| Q1 repetition (HL $\leq$ 7.8) | 0.4060 | High repetition |
| Q2 repetition | 0.3429 | Medium repetition |
| Q3 repetition | 0.4017 | High repetition |
| Q4 repetition (HL $>$ 13.4) | 0.2695 | Low repetition |
| Cohen's $d$ (Q1–Q4) | 0.825 | Large effect |

*Table 22.* Relationship between prototype half-life and repetition score. Lower half-life prototypes exhibit substantially higher repetition.

The results show a negative association between half-life and repetition, with a highly significant Spearman correlation and a large effect size ($d = 0.825$) between the lowest and highest quartiles. This provides evidence that **prototypes with shorter half-lives encode more local, repetitive lexical patterns**, whereas longer half-life prototypes correspond to broader, less repetitive activation structure.

**L1 Sparsity Ratio.** To measure the degree of "winner–take–all" behavior among prototypes, we compute

$$S_l = \mathbb{E}_x \frac{\max_p |a_{l,p}(x)|}{\frac{1}{P} \sum_{p=1}^{P} |a_{l,p}(x)|}. \tag{4}$$

A high value indicates that a single prototype (or a small subset) dominates the activation mass, reflecting strong concentration and effective sparsity.

**Gini Coefficient.** To quantify the inequality of the activation distribution, we compute the Gini index

$$G_l = \mathbb{E}_x \frac{1}{P} \left( P + 1 - 2 \frac{\sum_{p=1}^{P} (P + 1 - p) a_{l,p}^{\uparrow}(x)}{\sum_{p=1}^{P} a_{l,p}(x)} \right), \tag{5}$$

where $a_{l,p}^{\uparrow}(x)$ are the activations sorted in increasing order. Low values correspond to uniform activation across prototypes, while high values indicate strong inequality and specialization.

**Entropy.** To measure the spread or concentration of activations, we normalize $p_{l,p}(x) = |a_{l,p}(x)| / \sum_{q=1}^{P} |a_{l,q}(x)|$ and compute the Shannon entropy

$$H_l = -\mathbb{E}_x \sum_{p=1}^{P} p_{l,p}(x) \log p_{l,p}(x). \tag{6}$$

High entropy indicates diffuse activation across many prototypes, whereas lower entropy reflects concentrated, low-uncertainty routing.

**Mutual Information.** To assess how strongly prototype activations depend on surface lexical identity, we compute the mutual information between the discretized activations $\tilde{a}_{l,p}$ and the token identity $T$:

$$I_l = I(T; \tilde{a}_{l,p}). \tag{7}$$

High mutual information indicates that activations are predictive of the specific token type. A decrease in mutual information with depth does *not* by itself establish that deeper prototypes encode "more abstract" concepts; however, it is *consistent* with the broader pattern observed across our sparsity, entropy, and interpretability analyses, where later layers appear less tied to local lexical identity and more shaped by contextual or compositional signals.

Together, these metrics provide a multifaceted view of how prototype representations of ProtoT sharpen, specialize, and suggest a transition from local lexical cues to increasingly structured or context-sensitive behaviors as depth increases. LLaMA metrics by contrast, do not show clear systematic patterns for entropy, that remains uniform across layers and for Gini coefficient and L1 sparsity, that present an oscillating behavior. Mutual Information follows a pattern similar to that of ProtoT.

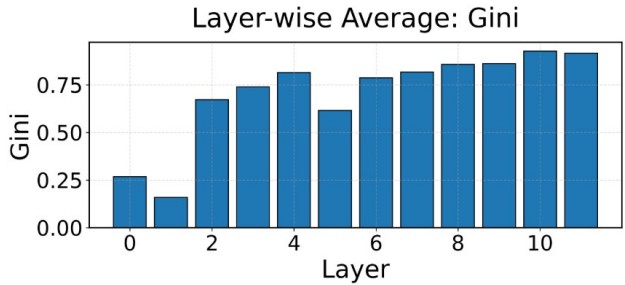

*(a)* Gini increases with depth, indicating concentration of activation mass.

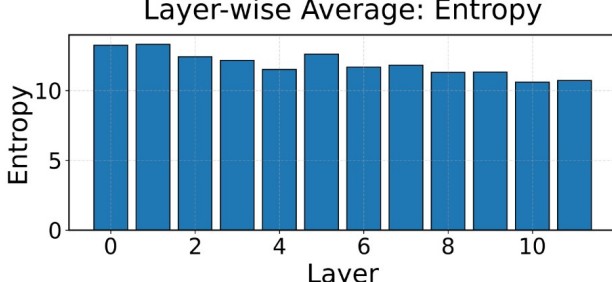

*(b)* Entropy decreases as routing becomes sharper and less uniform.

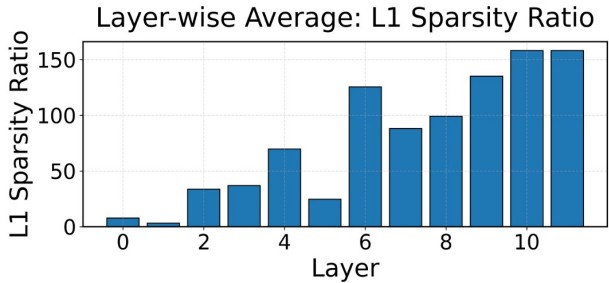

*(c)* L1 sparsity increases, reflecting increasingly selective activation.

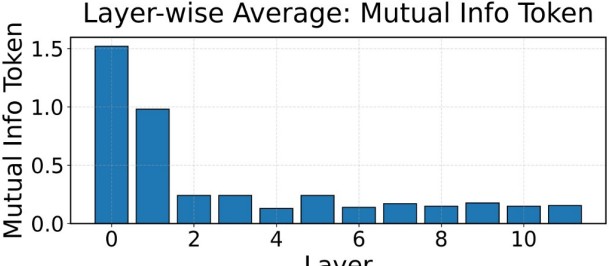

*(d)* Mutual information decreases with depth, indicating weaker dependence on lexical identity and greater sensitivity to contextual structure.

*Figure 7.* ProtoT interpretability metrics across depth.

## A.8. Additional Concepts Visualization and LLM-aided Evaluation Results

This section contains additional visualizations and randomly selected samples from the LLM scoring process. It also contains result statistics for LLM-aided evaluation experiment for multiple model configuration of ProtoT and for LLaMA.

**Prototype visualizations**  We provide additional examples from the write gate activation interpretability experiment, useful to better illustrate results about learned concept representation (figures 9, 10, 11, 12, 13).

**LLM scoring and labeling**  We provide visualizations of some random selected examples (Figures 14, 15, 16, 17.) and resulting statistics of the LLM scoring process. We also show ablation for different model configurations, including R=16, R=64, two extra seeds for R=32 and R=32 without low rank projection. (Figures 18, 19, 20, 21. 22, 23)

**Additional details and results of LLaMA SAE experiment**  We evaluate two additional baselines. For the SAE baseline, we train a sparse autoencoder on the residual stream of each LLaMA layer using 100M tokens and a dictionary size of 4,096, then select 32 features per layer via either highest activation variance or highest activation frequency, deliberately favoring the SAE by cherry-picking its most informative features while evaluating all ProtoT prototypes indiscriminately. For each selected feature, we identify the 10 most activating sentences and apply our LLM-aided scoring pipeline. For the null model baseline, we sample random subsets of 10 sentences from FineWeb, one per prototype (32 prototypes × 12 layers = 384 total), and apply the same scoring procedure to establish a lower bound on pipeline performance independent of any learned structure.

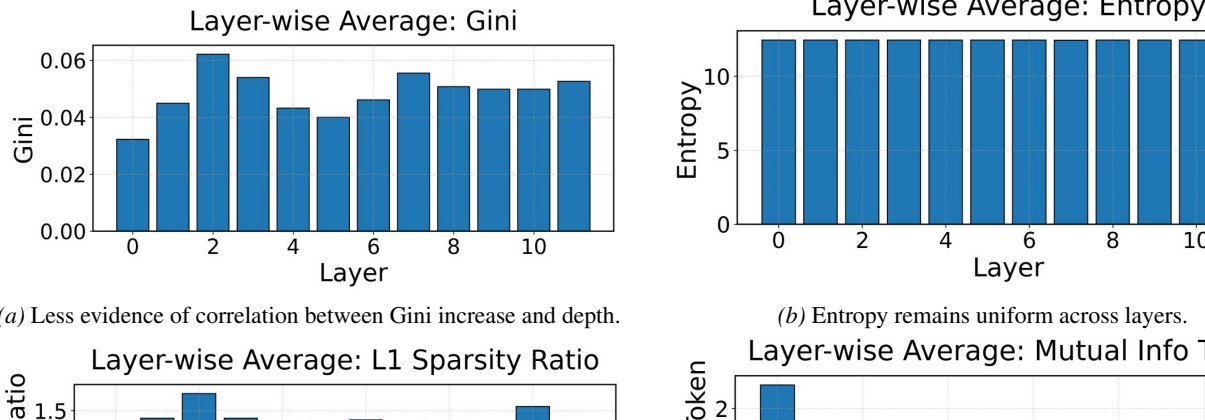

*(a)* Less evidence of correlation between Gini increase and depth.

*(b)* Entropy remains uniform across layers.

*(c)* L1 sparsity does not increase with depth.

*(d)* Mutual information shows a similar pattern in ProtoT and LLama.

*Figure 8.* LLaMA interpretability metrics across depth.

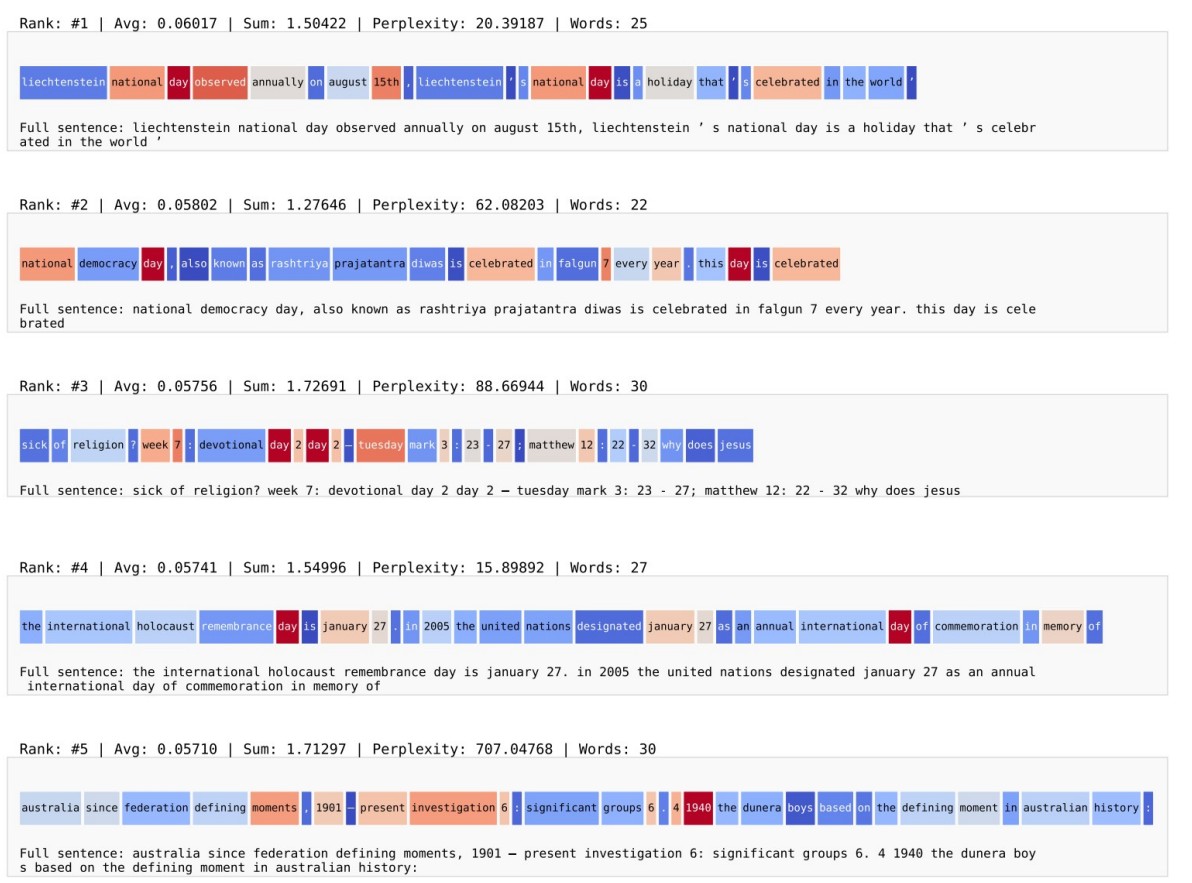

*Figure 9.* Visualization for prototype **L0 P18**. Half-life = 12.8.

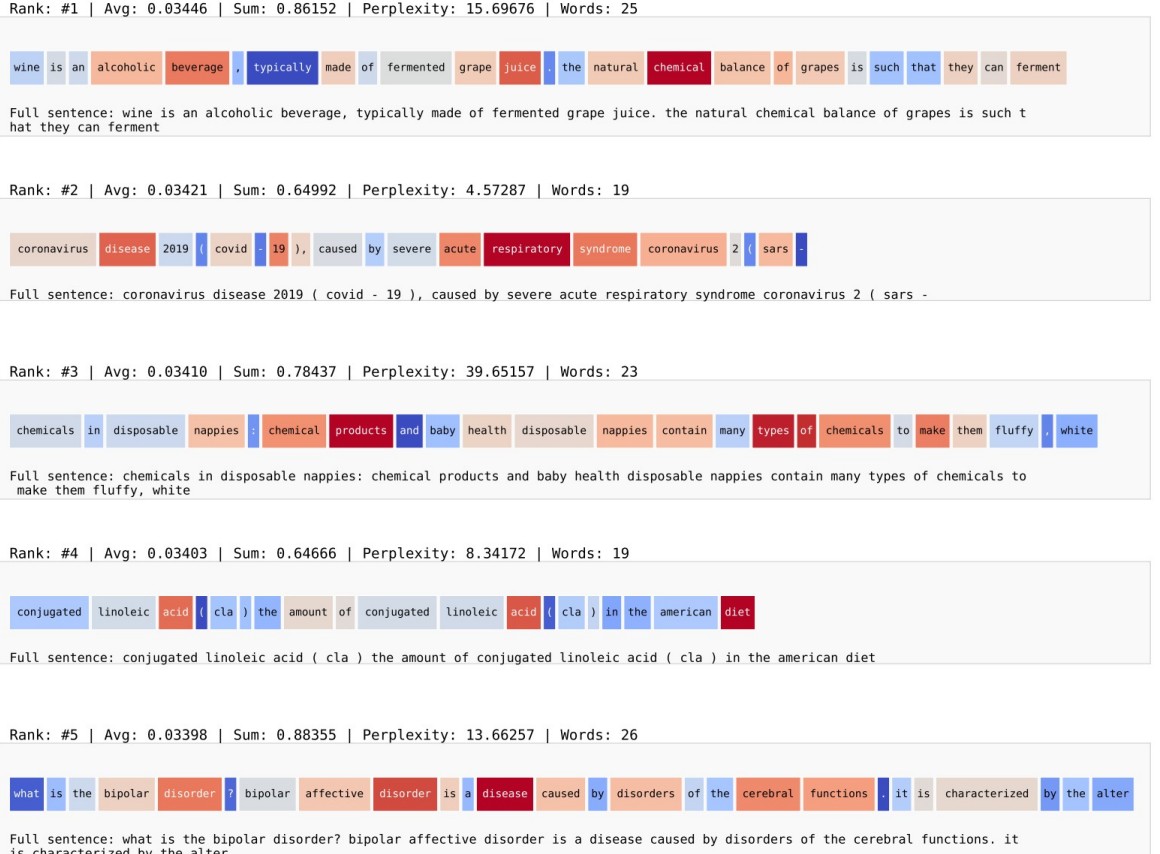

*Figure 10.* Visualization for prototype **L1 P14**. Half-life = 13.2.

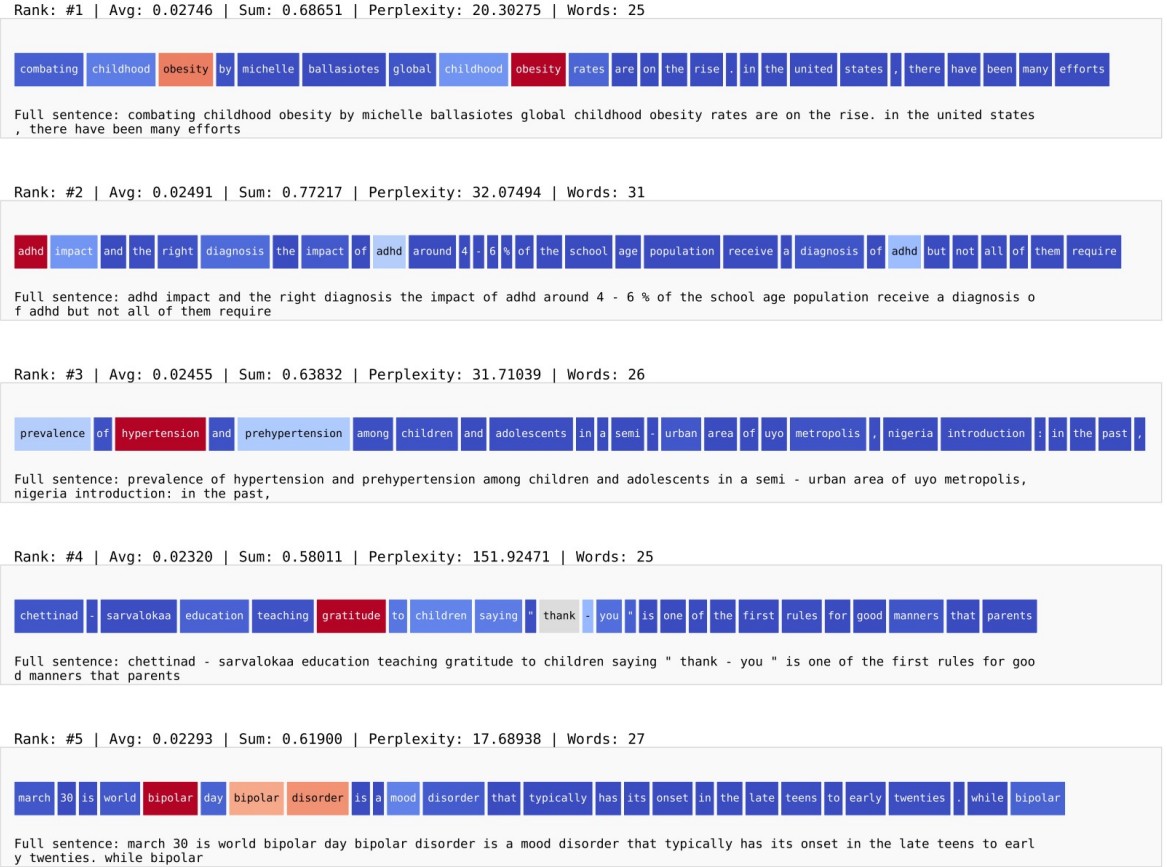

*Figure 11.* Visualization for prototype **L7 P31**. Half-life = 12.7.

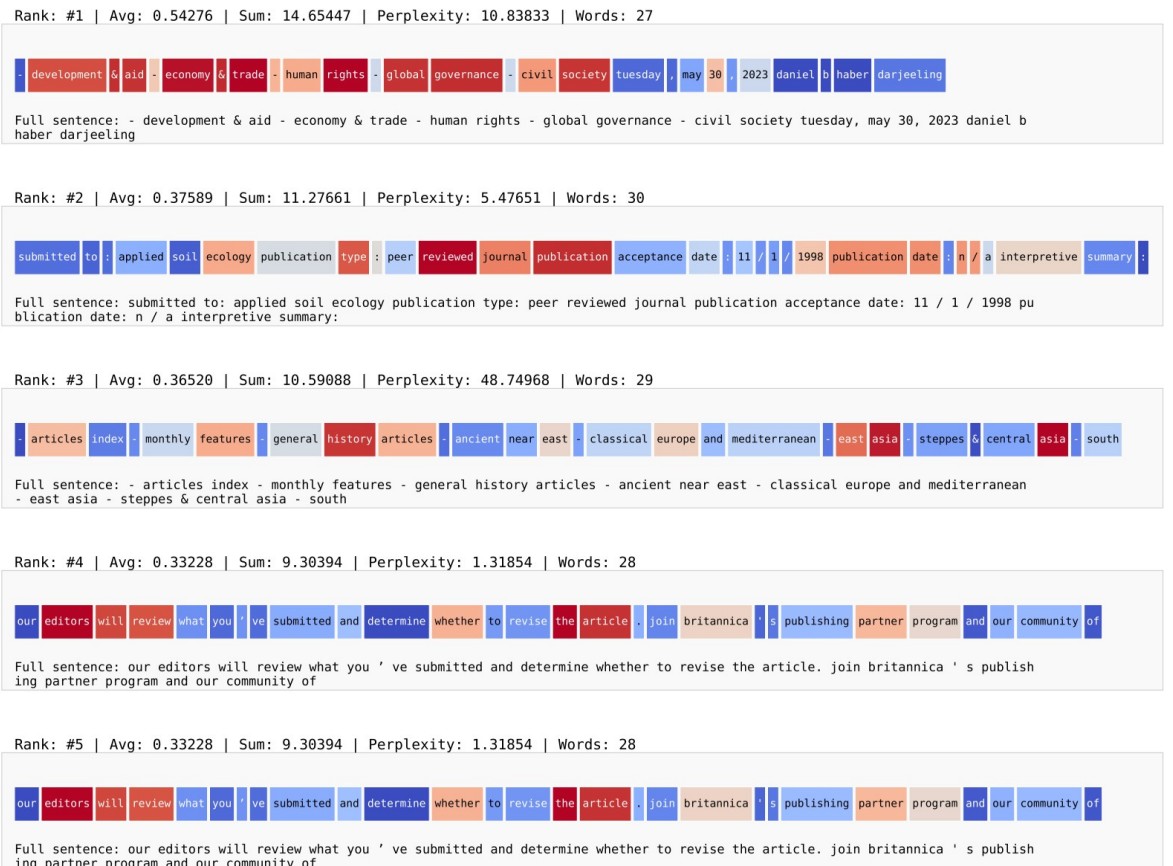

*Figure 12.* Visualization for prototype **L8 P5**. Half-life = 0.140.

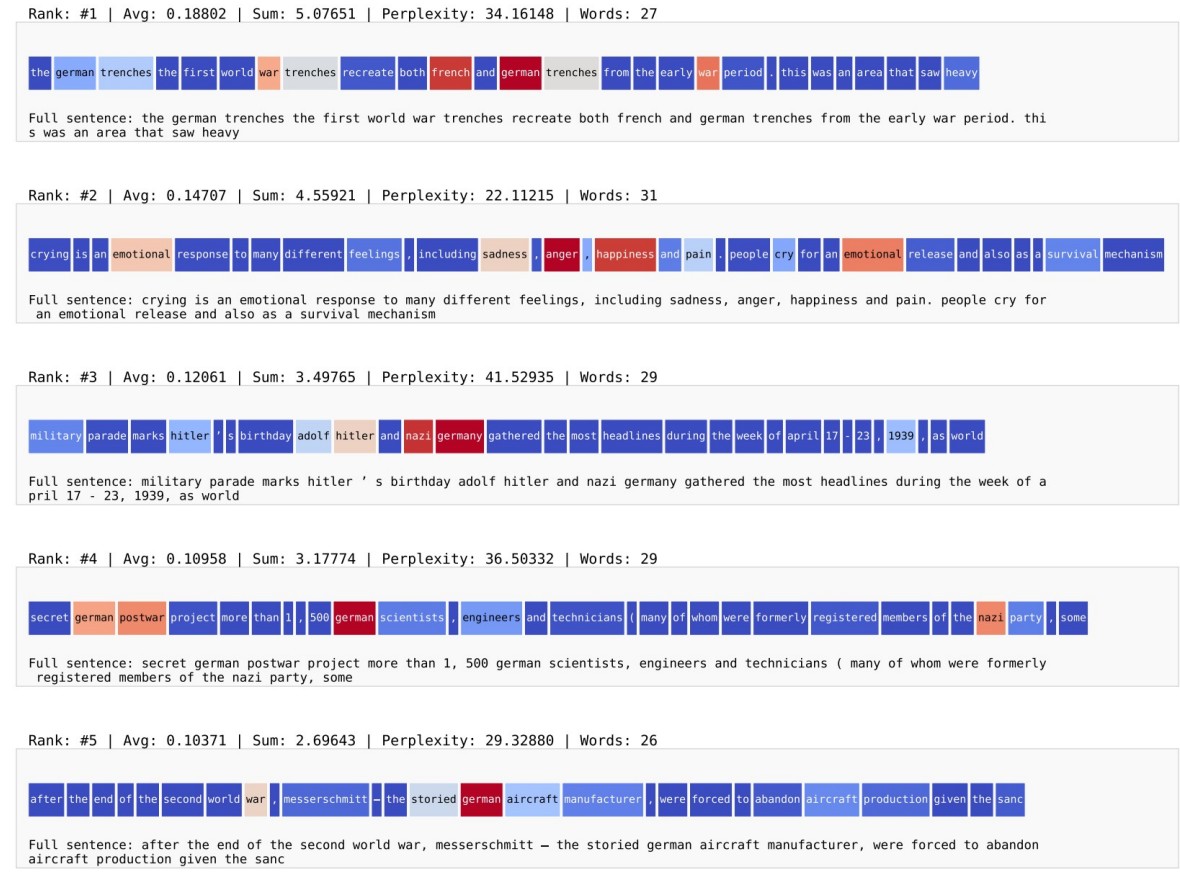

*Figure 13.* Visualization for prototype **L10 P8**. Half-life = 0.510.

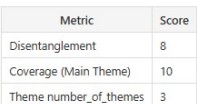

**Prototype L0 / P21**

| Metric | Score |
|---|---|
| Disentanglement | 8 |
| Coverage (Main Theme) | 10 |
| Theme number_of_themes | 3 |

**Main Theme:** negative contractions with auxiliary verbs (don't/doesn't/can't/isn't/aren't/couldn't/didn't)

**Explanation:** Across all ten sentences, the most activating tokens are negative contractions or closely related negative verb forms like "don't," "can't," "isn't," "aren't," "couldn't," and "didn't." This pattern is already clear from the token lists and is reinforced by the sentences themselves. Every example contains at least one such negative auxiliary construction near the peak-activation tokens, so coverage is maximal. Secondary variation includes different topics (politics, school, finances, science, parenting) and some punctuation or quotation context, but these are not consistent enough to constitute major themes. Thus, the main feature is strongly focused on negative contracted auxiliaries, with high coverage and only a few minor, less relevant themes, yielding a strong overall disentanglement score.

▼ View sentence snippets used for analysis

- sentence[1]: don t don t control over
- sentence[2]: lots students don t students students don t
- sentence[3]: there shootings not don t shootings shootings
- sentence[4]: ve heard don t isn t but
- sentence[5]: all bees tenor just isn t
- sentence[6]: power can didn t attractions
- sentence[7]: can t don t enough protege
- sentence[8]: ever had left just couldn t
- sentence[9]: anything don t struggle there crying
- sentence[10]: aren t quite classroom disrupt cause

▼ View full sentences

1. **Rank 1:** group 1: libertarians libertarians are people who don ' t want rulers and don ' t want other people to have control over them or responsibility
2. **Rank 2:** writing a law assignment is something that lots of students don ' t enjoy, especially first - year students. it happens because students don ' t have familiarity
3. **Rank 3:** there have been 13 mass school shootings since 1966, not 27 this year don ' t conflate mass shootings with school shootings. for many people
4. **Rank 4:** you ' ve likely heard the concept: don ' t work for money, make it work for you. this concept isn ' t new, but compound interest
5. **Rank 5:** i hear all the time that bees are vanishing — but are they really? the apocalyptic tenor of the rhetoric just isn ' t appropriate
6. **Rank 6:** bird power — ostriches can prove pretty ornery, but that didn ' t keep them from being displayed in some offbeat attractions. during
7. **Rank 7:** " if you can ' t explain it simply, you don ' t understand it well enough."- albert einstein the protege effect is a well -
8. **Rank 8:** have you ever had a dream that left an impression but when you woke up the next morning you just couldn ' t remember it? you know that it
9. **Rank 9:** making children do anything that they don ' t want to can be a massive struggle for everyone involved. there ' s a lot of crying, yelling,
10. **Rank 10:** every class has a student or two ( or five ) that aren ' t quite on board with your classroom agenda. they disrupt class, cause trouble,

*Figure 14.* LLM aided interpretability results for prototype **L0 P21** R=32 (S=135).

**Prototype L1 / P27**

| Metric | Score |
| --- | --- |
| Disentanglement | 6 |
| Coverage (Main Theme) | 9 |
| Theme number_of_themes | 3 |

**Main Theme:** descriptions of significant risks, problems, or stressors and their impacts in various domains

**Explanation:** Across almost all sentences, the activating tokens highlight situations involving threats, stressors, or problematic conditions (environmental stress on crops, fire emergencies, performance issues in manufacturing, risk management in business, earthquakes, addiction crises, social turmoil in the 1960s, vaccine shortages). Sentence 5 fits as well by focusing on studied relationships in an educational/clinical context, which is somewhat adjacent but still about conditions with important consequences. Sentence 3 is the least clearly about risk but still centers on a potentially significant artifact. The main pattern is not tied to a specific topic area but to the semantic field of hazards/problems and their effects. There are a few secondary themes (scientific/technical exposition, time-period or event descriptions), so the representation is not perfectly clean, but the risk/problem motif is clearly dominant in 8–9 of the 10 examples. Hence a moderate-to-strong disentanglement score, high coverage of the main theme, and a small number of overlapping secondary themes.

▼ View sentence snippets used for analysis

- sentence[1]: plants are exposed to several stresses
- sentence[2]: fire emergency alert australia ' combined perfect
- sentence[3]: ring that might have behind '
- sentence[4]: wire bonding in tends to
- sentence[5]: studies demonstrated a positive relationship between
- sentence[6]: management is something that has in '
- sentence[7]: earthquakes are moderate of ) magnitude
- sentence[8]: working in recovery has witnessed
- sentence[9]: 1960s had share ups downs lows
- sentence[10]: not enough for shots will be to

▼ View full sentences

1. **Rank 1:** crop plants are exposed to several environmental stresses, which all affect plant growth and development and consequently hamper the productivity of crop plants. drought is considered to be
2. **Rank 2:** australian fire emergency alert australia ' s record high temperatures combined with record low rainfall are a perfect ( fire ) storm. rivers are either dry or at record lows
3. **Rank 3:** a ring that might have been the inspiration behind jrr tolkien ' s ' lord of the rings ' and ' the hobbit ' books has
4. **Rank 4:** traditional wire bonding used in the semiconductor manufacturing industry tends to have performance related issues that are tied to inductance and capacitance. flip
5. **Rank 5:** previous studies demonstrated a positive relationship between deaf children ' s asl acquisition and their english literacy skills and the importance of parental language input. this study examined the
6. **Rank 6:** risk management is something that has to be given special attention in today ' s complex business environment. the market in which the businesses operate today is totally different to
7. **Rank 7:** glacial earthquakes are moderate earthquakes of ( surface - wave ) magnitude up to 5 on the richter scale. they are closely related to ice motion. they occur
8. **Rank 8:** anyone working in the field of addiction and recovery has witnessed firsthand the devastating effects of america ' s dependence on opioid narcotics.
9. **Rank 9:** the 1960s had its share of ups and downs, highs and lows. it was marred by protests, fights for injustice, the vietnam war,
10. **Rank 10:** still not enough for entire population, shots will be restricted to high risk only november 10, 2004 state and local health officials and the cdc have worked together to

*Figure 15.* LLM aided interpretability results for prototype **L1 P27** R=32 (S=135).

**Prototype L5 / P7**

| Metric | Score |
| --- | --- |
| Disentanglement | 4 |
| Coverage (Main Theme) | 4 |
| Theme number_of_themes | 7 |

**Main Theme:** educational or expository texts presenting facts, questions, or explanations

**Explanation:** Several sentences are explicitly educational or expository: Sentence 2 lists "five fun facts"; Sentence 7 is about a school board class guide; Sentence 8 outlines a Bible lesson with main point and key passage; Sentence 10 poses a history exam-style prompt. Others (3, 4, 5, 6) are informational research or academic-style descriptions, which loosely fit the same expository/educational flavor. However, there is notable heterogeneity: classical rhetoric (1), numbered fun facts (2), paleontology (3), engineering colleagues (4), economics inscription (5), plant research (6), school textbook (7), religious teaching guide (8), devotional reflection (9), and a history assignment (10). This diversity suggests multiple overlapping themes: academic/research context, religious instruction, historical or exam-like questions, numbered fact lists, and named professionals. The main theme of educational or explanatory discourse is present in roughly 4–6 sentences strongly and others more weakly, giving a moderate but not dominant pattern. Hence coverage_main_theme is 4 and number_of_themes is relatively high at 7, yielding a low-to-moderate disentanglement score of 4.

▼ View sentence snippets used for analysis

- sentence[1]: of there [...] secondly , persuasion
- sentence[2]: five fun facts : 1 . 2
- sentence[3]: ancient half continue says dr .
- sentence[4]: the colleagues dr soper dr mike
- sentence[5]: irving fisher professor fisher irving fisher
- sentence[6]: striped , researcher jennifer blake was
- sentence[7]: back : class 6th table -
- sentence[8]: bible : key : mark question
- sentence[9]: jesus . . elli , it
- sentence[10]: prove give . october | education

▼ View full sentences

1. **Rank 1:** of the modes of persuasion furnished by the spoken word there are three kinds. [...] secondly, persuasion may come through the hearers, when the
2. **Rank 2:** five fun facts about the irish holiday: 1. st. patrick was not born in ireland, but in britain. 2. green was not always the color
3. **Rank 3:** ancient crocodile swim traces found in tumbler ridge half a decade ago continue to contribute to global research of the reptiles, says paleontologist dr.
4. **Rank 4:** the measurements reported in this post were made by colleagues of the school of engineering at the university of birmingham – dr david soper and dr mike jesson –
5. **Rank 5:** inscribed by irving fisher to his famous economics rival, professor frank a. fetter fisher, irving and fisher, herbert w. constructive income taxation. a
6. **Rank 6:** striped maples wait to last minute before choosing their sex a few years ago, rutgers researcher jennifer blake - mahmud was working on a bot
7. **Rank 7:** back to: karnataka board class 6th english guide and notes table of contents - madhav: a father of a young child who takes his son
8. **Rank 8:** bible passage: luke 15 main point: jesus is the one who seeks and saves the lost. key passage: mark 6: 34 big picture question:
9. **Rank 9:** jesus was no stranger... by elli atchison during lent and holy week, it is appropriate for us to reflect on places in today ' s
10. **Rank 10:** prove that in the time of nero, no one was protected from the emperor ' s arbitrariness. give examples. october 31, 2020 | education

*Figure 16.* LLM aided interpretability results for prototype **L5 P7** R=32 (S=135).

**Prototype L9 / P14**

| Metric | Score |
|---|---|
| Disentanglement | 9 |
| Coverage (Main Theme) | 10 |
| Theme number_of_themes | 2 |

**Main Theme:** Temporal expressions specifying historical periods, date ranges, and years (often with prepositions like during/from/in)

**Explanation:** All ten examples center on temporal expressions: specific years, year ranges, and periods such as 'late 19th and early 20th centuries,' 'from July 4, 1941 to 1943,' and '1632–1723.' The most activating tokens consistently include prepositions plus time markers (during, from, in, till) tied to explicit dates or bounded time spans. Sentences cover historical events, movements, or seasons, but those topics are secondary to the consistent temporal-range pattern. There is a minor secondary theme of historical description, but it is tightly bound to the use of precise time references. Because nearly every trigger is a time phrase, the coverage is maximal and the number of distinct themes is low. This yields a highly, though not perfectly, clean temporal-range feature, justifying a disentanglement score of 9.

▼ View sentence snippets used for analysis

- `sentence[1]: during late 19th early 20th`
- `sentence[2]: from july 4 , 1941 to`
- `sentence[3]: during 17 - 23 , 1939`
- `sentence[4]: presidential in the states first in`
- `sentence[5]: in from the early 1940s to`
- `sentence[6]: of the of the early 20th`
- `sentence[7]: with season from july till in`
- `sentence[8]: has been this to the 2018`
- `sentence[9]: of the last few hundreds of`
- `sentence[10]: 1632 - 1723 in`

▼ View full sentences

1. **Rank 1:** emanuel jules joseph descomps was a french sculptor and jeweller working in paris during the late 19th and early 20th centuries. known as
2. **Rank 2:** pinsk ( belarus ) was under soviet rule for seven months and under the german occupation from july 4, 1941 to 1943. at the start
3. **Rank 3:** military parade marks hitler ' s birthday adolf hitler and nazi germany gathered the most headlines during the week of april 17 - 23, 1939, as world
4. **Rank 4:** 1792 presidential election the united states presidential election of 1792 was the second presidential election in the united states, and the first in which each of the original
5. **Rank 5:** mccarthyism is a term describing the intense anti - communist suspicion in the united states in a period that lasted roughly from the early 1940s to
6. **Rank 6:** the anti - sexual violence movement has a rich history, one that began long before the feminist movement of the 1970s or the suffragists of the early 20th
7. **Rank 7:** the gambia has a tropical climate, with a hot and rainy season from july till october and the dry period, in which cooler temperatures predominate.
8. **Rank 8:** as the u. s. flu season has been churning away, with nearly double the number of pediatric deaths at this point compared to the 2018 flu
9. **Rank 9:** dodo – the emblem of extinction the dodo is probably the most famous in the long line of extinct animals of the last few hundreds of years.
10. **Rank 10:** anton van leeuwenhoek ( 1632 - 1723 ) was a tradesman and scientist from delft, in the netherlands.

*Figure 17.* LLM aided interpretability results for prototype **L9 P14** R=32 (S=135).

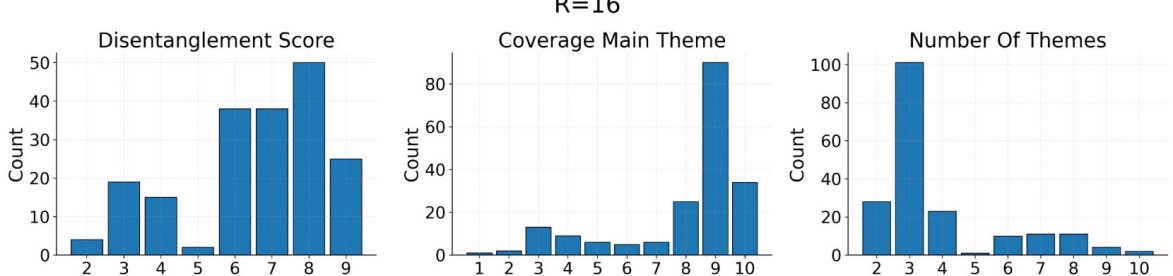

*Figure 18.* histograms for LLM aided interpretability for model configuration R=16.

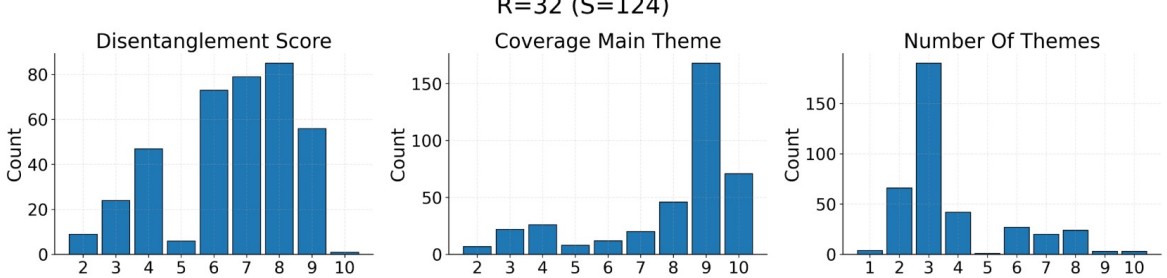

*Figure 19.* histograms for LLM aided interpretability for model configuration R=32 (S=124).

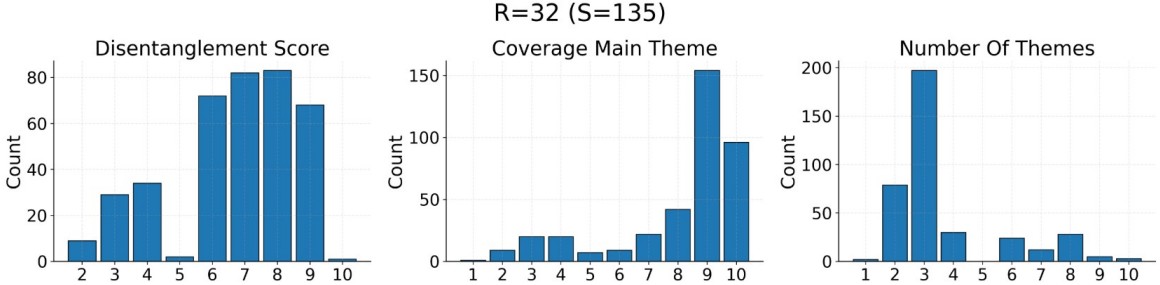

*Figure 20.* histograms for LLM aided interpretability for model configuration R=32 (S=135).

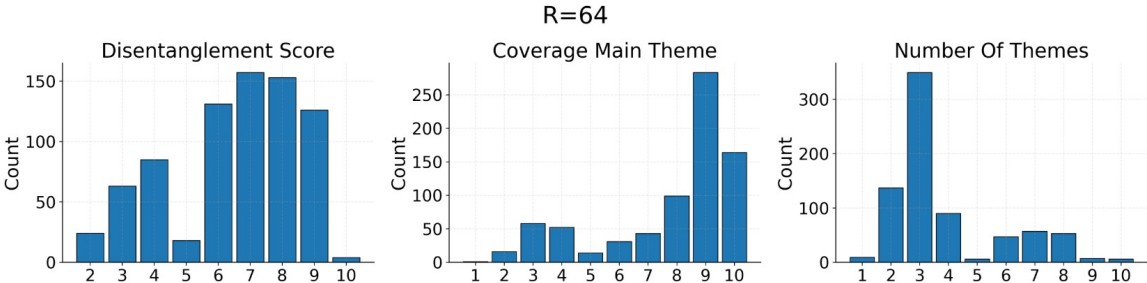

*Figure 21.* histograms for LLM aided interpretability for model configuration R=64.

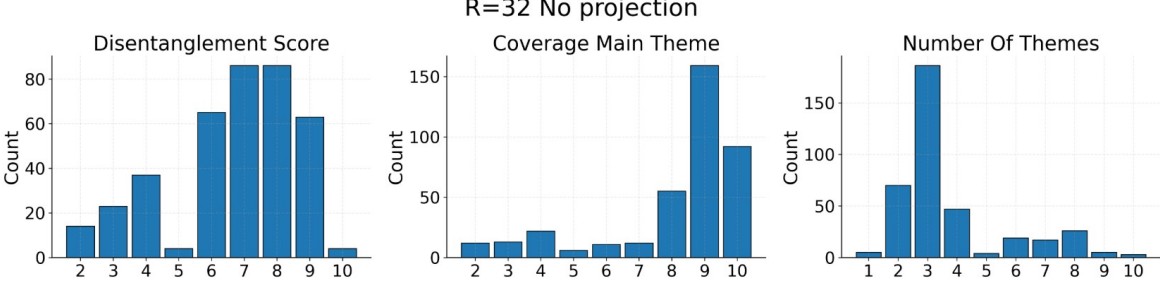

*Figure 22.* histograms for LLM aided interpretability for R=32 (S=124) and no low rank projection.

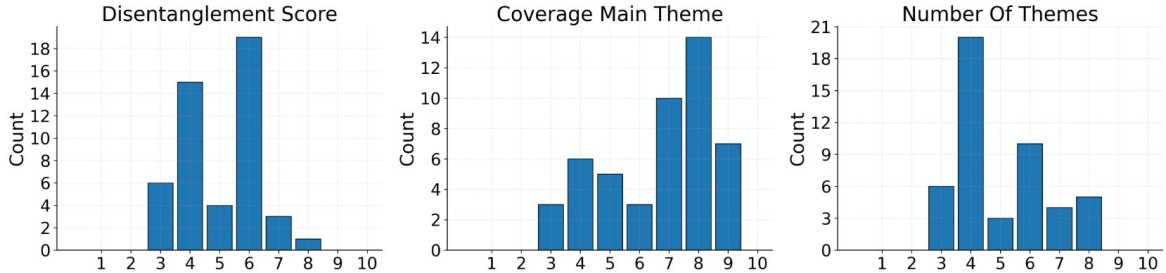

*Figure 23.* histograms for LLM aided interpretability for LLaMA model trained from Table 1.

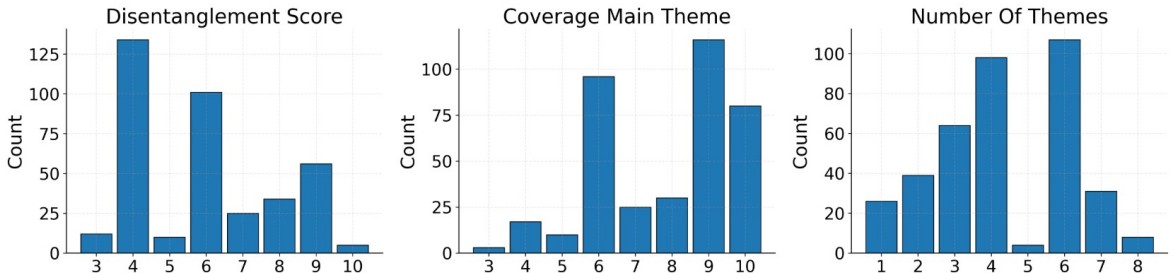

*Figure 24.* histograms for SAE, selected by highest activation variance.

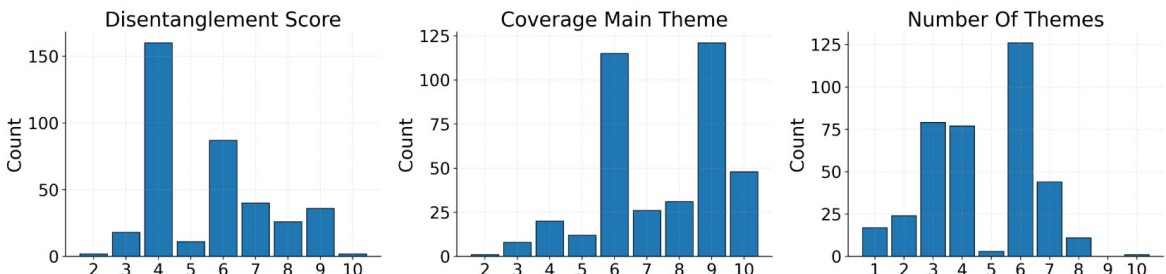

*Figure 25.* histograms for SAE, selected by highest activation frequency.

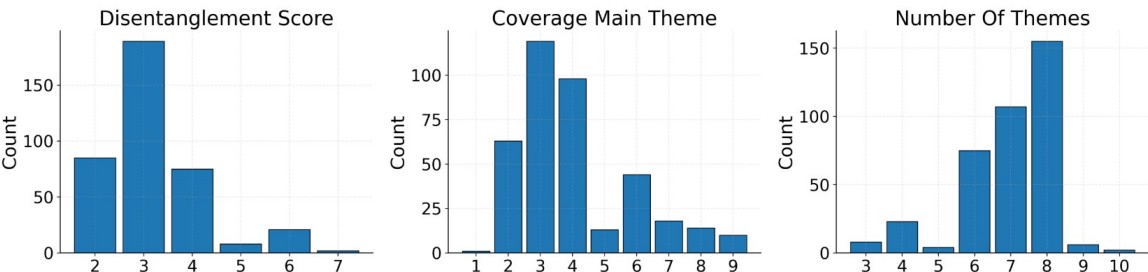

*Figure 26.* histograms for null model baseline.

## A.9. Prompt for LLM-aided interpretability experiment

```
        "You are analyzing a single prototype (a neuron-like feature) from a neural
language model.\n"
        "For this prototype you are given, for each of its top-ranked sentences, the
full sentence and the subset "
        "of its most activating tokens. Each example is formatted as:\n\n"
        "  Most activating tokens sentence: <token1 token2 ...>\n\n"
        "  Sentence: <full sentence text>\n"
        "A **theme** is any recurrent characteristic that appears across multiple
high-activation token sets or their "
        "sentences. Themes can be narrative motifs, entities, stylistic elements,
punctuation patterns, lexical fields, "
        "or any other shared property that appears across more than one example. ALL
PREVIOUS EXAMPLES ARE CONSIDERED THEMES. It can be local (individuated by single"
        "isolated words (like numbers or punctuation) or more sentence level
(individuated by composing multiple tokens)\n\n"

        "Your task is to determine whether there is a meaningful main theme shared
across the provided sentences, "
        "and how strongly that theme characterizes this prototype.\n\n"

        "You should first of all observe the most activating tokens to check if there
is a main theme that is already observable there"
        "then if by observing only the tokens it is hard to find a common theme, you
should check the sentences to observe if there is"
        "a more 'sentence level' theme like a narrative or a motif. Remember that the
first indication is given by the most activating tokens,"
        "if they are clearly forming a visible coherent pattern, that's enough to
conclude the analysis. If instead the pattern they are forming is not that clear,"
        "then move to observe and analyze the full sentences."

        "For example in these sentences and most activating tokens there is everywhere
a common theme of comitative structure with the use of the word 'with' that is already
very visible with the most activating tokens"
        "Sentence Rank 1: how are nigeria â\200\231 s trading and political
relationships changing? what relationship did nigeria have with britain? trade between
britain and west africa has occurred for over"
        "Sentence Rank 2: the teens at risk series deals with powerful and relevant
issues facing teenagers today. this series reveals first - hand how young people deal with
and overcome the very real"
        "Sentence Rank 3: the only country that south korea shares a land border with
is north korea. the land border between the two countries is 148 miles in length, dissec"
        "Sentence Rank 4: communication with families: a plan for every child keep
communicating with parents and pave the way for continued progress! by carla poole summer
is on the"
        "Most activating tokens[1]: trading relationships relationship with trade
between"
        "Most activating tokens[2]: deals with issues facing deal with overcome"
        "Most activating tokens[3]: shares border with border between two"
        "Most activating tokens[4]: communication with families communicating with
parents"

        "or in this other example list of most activating tokens, we have knowledge
and questions, this can be safely identified as a single common main theme. the list of
sentences reinforce what was already observable"
        "Sentence Rank 1: did you know that something called chronic fatigue existed?
well, many of us do not know about this ailment, but it is actually quite common. the"
        "Sentence Rank 2: how to do homework quicker the best way to do homework
sooner is a problem most college students have. doing homework isn â\200\231 t as
enjoyable as mother and"
        "Sentence Rank 3: what does shingles look like? how does shingles start? how
does shingles affect the body? how is the diagnosis made and treatment"
        "Sentence Rank 4: norovirus information guide noroviruses ( previously known
as norwalklike viruses ) are widely recognized as the agents known to cause outbreaks of
illness among"
        "Most activating tokens[1]: did know well do know quite"
        "Most activating tokens[2]: do do isn t as as"
        "Most activating tokens[3]: does ? does ? does ?"
        "Most activating tokens[4]: known as recognized as known"
```

```
        "In this other examples we have that even if the sentences don't exihibit a
clear evident patterns, the most activating tokens are all about punctuation. This is
enough"
        "to identify a hihgly disentangle punctuation theme with high coverage and low
number of themes."
        "Most activating tokens[1]: ; ; ; ; ; ; ; ; ; ; ;"
        "Most activating tokens[2]: , ), . . . , , ."
        "Most activating tokens[3]: , , , . . ,"
        "Most activating tokens[4]: . , , ,"

        "Sentence Rank 1: conflict resolution – for kids conflict; listening;
negotiate; compromise; resolve; resolution; understanding; working together; outcomes;
skill; listen; feelings; put"
        "Sentence Rank 2: schulman, samuel ( 1864 â\200\223 1955 ), u. s. reform
rabbi. schulman, born in russia, was taken to the u."
        "Sentence Rank 3: we provide arts and architecture assignment help in fine
arts, visual arts, industrial design, architectural design etc. at myassignmenthelp. net,"
        "Sentence Rank 4: google. com, pub – 6983070578797176, direct, f08c47f"

        "Break the disentanglement assessment into these components (all on a
1â\200\22310 scale):\n"
        " – coverage_main_theme: in how many sentences the main theme appears?\n"
        "    indicate the precise number between 1 and 10\n\n"
        " – number_of_themes: how many uncorrelated themes appear in the prototype?\n"
        "    indicate the precise number between 1 and 10, if it is more than 10,
just answer 10\n\n"

        "Use these two component scores to decide the overall disentanglement_score
(1â\200\22310):\n"
        " 1–2 = There is no recurring characteristic across the most activating
tokens or the sentences; entirely mixed or noisy.\n"
        " 3–4 = Very weak hints of a pattern; mostly mixed, but with minor
repetition.\n"
        " 5–6 = A moderate theme: some noise, but a noticeable dominant trait.\n"
        " 7–8 = A strong theme: clearly recurrent and consistent across many
sentences.\n"
        " 9–10 = An extremely strong and clean theme: nearly all sentences share the
same core characteristic.\n\n"

        "When choosing the scores, do NOT avoid extremes: use the full range from 1 to
10 whenever appropriate.\n\n"

        "Provide your answer STRICTLY as a JSON object with these exact keys:\n"
        ' "disentanglement_score": integer 1–10,\n'
        ' "coverage_main_theme": integer 1–10,\n'
        ' "number_of_themes": integer 1–10,\n'
        ' "theme": short string describing the single main shared characteristic,\n'
        ' "explanation": 1–10 sentences explaining why you chose these scores.\n\n'
        "Output ONLY valid JSON. No markdown, no backticks, no additional text."
```

*Figure 27.* Full prompt used for LLM aided evaluation and labeling experiment

# B. Appendix: Comprehensive Evaluation Details

## B.1. Qualitative Evaluation Methodology

We adopted an LLM-as-a-judge protocol inspired by Chatbot Arena (Chiang et al., 2024). For a given evaluation prompt, we present the two model outputs (Response A and Response B) to a frozen judge model (Gemma-3-4B-IT) using a fixed system instruction that asks the judge to select the better response based on coherence, relevance, fluency, and correctness, and to reply with a single token in "A", "B", "Tie". To reduce position bias, we query the judge twice per example: once with the order (A, B) and once with the order (B, A). The two decisions are mapped back to the original models and converted into soft pairwise scores (win = 1, tie = 0.5 per model). Aggregating over all prompts yields pairwise win/tie statistics between the two systems, which we then use to compute Elo ratings following the standard Chatbot Arena procedure. The evaluation prompts used for scoring are 100 randomly chosen prompts from the FineWeb-test set, resulting in a total of 600 pairs of comparisons for the LLM judge.

## B.2. Qualitative Samples

We decode 50-token continuations for each model on a shared set of FineWeb validation prompts, score every output with BLEU against the held-out reference, and then extract the highest-BLEU prompt for each model. The examples below present those prompts alongside every model's completion and BLEU score, enabling apples-to-apples qualitative inspection.

Each subsection lists the prompts where a model achieved its highest BLEU scores. For every selected prompt, we show the prompt, the reference completion, and the completions (with BLEU) for all available models.

### B.2.1. PROMPT — PROTOT WINNER

**Prompt:** *..can also be caused by other conditions such as benign prostate enlargement. there are no known causes for prostate cancer. however, between 5% and 10% of cases run in families, where the patient inherits a high risk of this type of cancer. prostate cancer is very rare in men under 50. the risk increases after the age of 50 with half of all cases occurring in men over 70. men from families with a history of prostate cancer are at higher risk than normal. race also has*

**Reference:** *an effect: men of afro-caribbean descent are about twice as likely to get it whereas men of asian descent have a lower risk of prostate cancer. some evidence suggests that a diet high in tomatoes, vitamin e, cruciform vegetables (such as broccoli, cabbage, cauliflower and brussels sprouts) and selenium may reduce the risk of prostate cancer. however, other studies have failed to confirm these effects, so the findings of this prostate cancer research have*

**ProtoT (BLEU 0.0359):** a higher risk of prostate cancer. the risk of prostate cancer is higher in men than women. the risk of prostate cancer is higher in men than women. the risk of prostate cancer is higher in men than in women. the risk of prostate cancer is

**DeltaNet (BLEU 0.0334):** a higher risk of developing prostate cancer. the risk of prostate cancer is higher in women with a high risk of developing the disease. the risk of prostate cancer is higher in women with a high risk of developing the disease. the risk of prostate cancer is

**LLaMA (BLEU 0.0000):** a risk. men from family with a history of prostate cancer are at higher risk than normal. men from family with a history of prostate cancer are at higher risk than normal. men from family with a history of prostate cancer are at higher risk than normal

**Mamba (BLEU 0.0000):** a significant impact on the risk of developing prostate cancer. african american men are at a higher risk of developing prostate cancer than caucasian men. african american men are at a higher risk of developing prostate cancer than caucasian men. caucasian men are at

### B.2.2. PROMPT — LLAMA WINNER

**Prompt:** *.. grow out of control to form a mass or tumour. it is a very variable disease. some tumours remain small and grow so slowly that they cause no problems for the rest of a man's life; others are aggressive, grow quickly and become life-threatening. many of these aggressive cases will eventually spread to the bones, where they can cause severe pain.*

*prostate cancer is now the most commonly diagnosed male cancer in many western countries. currently there are 37,000 new cases a year*

**Reference:** *in the uk; 218,000 in the usa; 25,000 in canada and 17,000 new cases a year in australia. the number of recorded cases has increased a lot in recent years. this is partly due to the increased use of the psa test, which has resulted in more cases being detected, and partly due to the fact that men are living longer. like most cancers, prostate cancer is more common in those over 60. main symptoms include: however, all of these symptoms*

**LLaMA (BLEU 0.0228):** there are more than 1,000 new cases a year. there are more than 1,000 new cases a year. there are more than 1,000 new cases a year. there are more than 1,000 new cases a year. there

**Mamba (BLEU 0.0203):** in the uk. the number of new cases is expected to rise to more than 1,000 by 2050. the disease is also becoming more common in the uk, with more than half of all new cases diagnosed in the uk in the last decade

**ProtoT (BLEU 0.0000):** in the united states. prostate cancer is the most common cancer in men and women. it is the most common cancer in men and women. it is the most common cancer in men and women. it is the most common cancer in men and women.

**DeltaNet (BLEU 0.0000):** in the u. the most common form of the disease is prostate cancer. the most common form of the disease is prostate cancer. the most common form of the disease is prostate cancer. the most common form of the disease is prostate cancer. the most

### B.2.3. PROMPT — DELTANET WINNER

**Prompt:** *averages and comparatives – hotter, colder, more, less, the highest, the lowest and so on. the students also learnt the basic elements of the water cycle. the biology unit included learning the characteristics of things that are alive: movement, the ability to eat and drink, the ability to breathe, reproduction (plants only !!), growth and excretion. the students were introduced to the process of photosynthesis to illustrate the fact that plants make their own food and ex*

**Reference:** *crete oxygen and water. this was a completely new concept for them and they found it fascinating. the students also learnt how insects pollinate plants to create new seeds. the phonics programme used the multisensory jolly phonics system, which we have introduced to all our beginner classes. each letter sound has an accompanying action. students and teachers have fun miming out words and students discover that they are able to spell long words if they listen for individual sounds and that they can decode*

**DeltaNet (BLEU 0.0143):** crete it. the students also learned how to use the system to make their own food. the students also learned how to use the system to make their own food. the students also learned how to use the system to make their own food. the students

**ProtoT (BLEU 0.0134):** cretion. the students also learned about the different types of plants and animals. the students also learned about the different types of plants and animals. the students also learned about the different types of plants and animals. the students also learned about the different types

**Mamba (BLEU 0.0114):** crete it. the students also learned about the process of photosynthesis and how it works. the students also learned about the process of photosynthesis and how it works. the students also learned about the process of photosynthesis and

**LLaMA (BLEU 0.0000):** eat them. the students learnt the importance of the importance of the water cycle to explain how it is important to the life of the plants. the students learnt the importance of the water cycle to explain how it is important to the life of the plants.

### B.2.4. PROMPT — MAMBA WINNER

**Prompt:** *babies that are smarter, taller or better athletes. hair and eye color could be manipulated. iqs could be boosted or lowered. it raises all kind of brave new world issues about genetically engineering the human race. moratorium gains momentum in the last week or so, there's been a flurry of statements from several groups of scientists warning about all this. mit's technology review had an in-depth report on the whole issue a couple of weeks back*

**Reference:** *, if you want to learn more. this week, groups that include the university of california's jennifer doudna, one of the researchers who developed crispr, essentially called for a moratorium on any attempt to do modification of the human germline using these techniques – at least until there's been more time for public discussion and more research to understand how well it works and how safe it is. in interviews, several of the scientists and bioethicists issuing these statements*

**Mamba (BLEU 0.0135):** the report, which was published in the journal science, was a response to a recent study by the university of california, berkeley, which found that the number of people who were genetically engineered to be genetically engineered was about 1 in 4.

**LLaMA (BLEU 0.0000):** the report was published in the journal science. the report was published in the journal science. the report was published in the journal science. the report was published in the journal science. the report was published in the journal science. the report was published

**ProtoT (BLEU 0.0000):** the report, "the future of science," was released on thursday. the report, which was released on thursday, was based on a study of the effects of the covid-19 pandemic on the human body. the study found that

**DeltaNet (BLEU 0.0000):** the report was published in the journal of the american society. the report was published in the journal of the american society. the report was published in the journal of the american society. the report was published in the journal of the american society. the

