# OpenReview forum: "Prototype Transformer: Towards Language Model Architectures Interpretable by Design"
_ICML.cc/2026/Conference — ICML 2026 regular_

### Official Review · Reviewer_QVR2 · 2026-03-06

**Soundness:** 3
**Presentation:** 4
**Significance:** 3
**Originality:** 3
**Overall Recommendation:** 4
**Confidence:** 3

**Summary:**

This paper introduces a prototype-based autoregressive LM architecture. The core idea is to incorporate prototype channel vectors that aggregates contextual information, instead of self-attention, allowing internal representations to be more interpretable, while also reducing quadric computational of attentions. The authors evaluate the architecture across several settings (language-modeling, GLUE, and generation quality with LLM-as-a-judge). They also conduct a series of interpretability experiments to understand the internal mechanisms and how information flows through the prototype structure. Overall, the results suggest that ProtoT maintains competitive performance on the evaluated tasks while providing a more accessible window for model interpretability.

**Compliance With Llm Reviewing Policy:**

Affirmed.

**Final Justification:**

I think the paper is good, and the rebuttal was also fine. I selected "(a) Fully resolved" because the other options are less suitable. I still have concerns that the rebuttal did not address, but I think they just stem from disagreeing with the authors. I don't have follow-up questions or "major concerns." My concerns are mostly about the correct framing of the architecture and the results. The claims in the paper should be softened (this is my view; the authors probably disagree). I still believe the paper is good, and the description of the overall score of 4 aligns perfectly with what I think of the paper.

**Key Questions For Authors:**

none

**Limitations:**

yes

**Strengths And Weaknesses:**

I find the architectural idea both novel and interesting, and I appreciate how the authors present and motivate it. In my opinion, pursuing self-explainable model architectures is somewhat of a holy grail for the field and is a direction worth exploring, while the architecture is not exactly "interpretable by design", it is in the right direction. The architecture offers linear computational scaling with sequence length, which is a valuable additional property. I was somewhat surprised to see that the architecture remains competitive to existing architeturs, and I found it quite impressive. The authors conduct a large number of experiments, and I appreciate the amount of work invested in the empirical evaluation. This is one of the rare cases where writing "conducted extensive experiments" feels justified. Finally, I also appreciate the overall presentation. The paper is well written and polished.

**Weaknesses:**

While the authors claim they propose an architecture that is interpretable by design, this is, in my opinion, false. It is in the right direction, but it is not interpretable. What they can say is that the architecture is "easier to interpret". I do not see any difference in the interpretation protocol of the prototypes compared to what is currently done in mechanistic interpretability (except, again, that it is easier with their architecture because there are fewer prototypes than neurons, for example). There is nothing "human-interpretable" here; in practice, they still need to analyze activations (of prototypes rather than neurons, attention heads, or other components) and search for something meaningful. This aligns completely with what is done in mechanistic interpretability for transformers (as far as I know).

The empirical competitiveness is somewhat mixed. On the one hand, it is encouraging that ProtoT achieves performance that is reasonably close to baselines despite replacing the standard attention mechanism. On the other hand, there remains a noticeable performance gap.In addition, the experiments are conducted on relatively small-scale models and datasets, and on datasets that are less relevant for 2026. I would expect more instruction-following datasets than GLUE, which is more relevant to encoder models (and not to autoregressive architectures). That said, in my opinion this is a relatively minor weakness. I do not expect a newly proposed architecture to outperform well-studied and highly optimized architectures such as transformers. Still, considering that the experimental setup was likely designed to support ProtoT, it is somewhat concerning that the method consistently lags behind the other architectures in the comparisons.

What I believe the authors should pursue, perhaps as future work or in the case the paper is rejected (I think it should be accepted), is moving closer to truly interpretable-by-design features. One possible direction would be to take a pre-trained ProtoT and develop ways to adjust or align certain prototypes so that they correspond to semantic roles that stakeholders actually care about. In this way, stakeholders using the model could better understand its behavior in the context of concepts that are meaningful to them. For example, in a medical domain, one could imagine adjusting some prototypes so that each corresponds to a different subgroup of symptoms. After fine-tuning the model on disease classification data, it would then be possible to analyze how these symptom-related prototypes behave and interact, potentially yielding useful insights about the domain. This is somewhat hypothetical direction (I have no idea if it can work), but in my view it reflects the real potential of ProtoT: enabling models whose internal structure can be **easily** aligned with human-relevant concepts -- but in a top-down approach.

---

> ### Author Rebuttal · Authors · 2026-03-31
>
> We thank the reviewer for their time and suggestions!
>
> ---
>
> > **Answer to Weakness 1 (“While the authors claim they propose an architecture that is interpretable by design, this is, in my opinion, false [...]”):**
>
> We address the two aspects of this statement, and we will communicate this better in the paper:
>
> 1. Whether we have achieved full interpretability. The answer is clearly No. We do not have full native transparency about what the model is thinking at every layer. The paper title (“Towards […]”), content (with phrases like “incorporating interpretability considerations”), and experiments, make it clear that we have made only partial progress to this ultimate goal.
>
> 2. Whether ProtoT’s design choices encourage interpretability beyond “fewer prototypes than neurons". The answer is Yes, and it goes beyond “easier to interpret”. As we explain at length in our answer to reviewer 6GP2, Weakness 2., ProtoT’s learning is naturally pressured toward monosemanticity and disentanglement, while in Transformer superposition arises naturally.
>
>  "they still need to analyze activations" - Sparse Autoencoder methods (SAE) require not only collecting activations but also training an auxiliary model for each component under inspection. ProtoT needs none of this: its activations are directly inspectable and achieve competitive monosemanticity with SAE-based methods out of the box (see next paragraph).
>
> While our goal is not to substitute Mechanistic Interpretability (MI) methods, to prove ProtoT’s  competitiveness, we trained a SAE for each layer of LLaMA and evaluated its features with the LLM scoring pipeline. The results show that ProtoT scores better across all dimensions. Details are in our answer to reviewer T9md (Cons 2. and Question (1/1)).
>
>
> ---
>
> **Answer to Weakness 2 (“The empirical competitiveness is somewhat mixed [...]”):**
>
> “there remains a noticeable performance gap” - we have answered this comprehensively in our response to Reviewer T9md, “Answer to Cons 1.”.
>
> “the method consistently lags behind the other architectures in the comparisons” - we disagree, most notably because ProtoT outperforms DeltaNet almost across the board. An exception are the Long-context scalability results, as ProtoT scales worse than the baselines, but in that experiment we also provided an explanation with empirical justification why that may be, and why it may not matter in practice. In short, the results suggest that the hidden size of ProtoT restricts its long-context scalability the most, and that when jointly scaling context, model size, and data size (in large-scale experiment), ProtoT shows competitive scalability, suggesting that in practice this may not be an issue.
>
> “The experimental setup was likely designed to support ProtoT.” - we firmly disagree with this statement, and we explain the choice of each dataset:
>
> 1. The choice of FineWeb as a pre-training corpus (and for test-evaluation of perplexity) is due to its properties as: realistic, varied, representative, and freely-available.
>
> 2. The choice of GLUE instead of instruction-following benchmarks:
>
>      2.1 It is a widely used, established NLU fine-tuning benchmark that still differentiates models at our scale and probes a broad range of abilities in pre-trained LMs, including autoregressive ones.
>
>      2.2 It complements, rather than replaces, our “Text-Generation Performance” experiment and provides a standardized downstream setting for comparing base architectures under a shared supervised fine-tuning recipe. In contrast, instruction-following benchmarks target a different capability and depend more strongly on the instruction-tuning recipe, while our goal here is to compare base architectures as directly as possible.
>
> 3. For the text-generation experiment, we updated the experimental setup to use a random 100-document subset of FineWeb-test, increased from 20. The choice of data source is natural as we train on FineWeb-train. The details and results are in our response to reviewer T9md, in “Answer to Cons 1.”, point 3.
>
> ---
>
> **Answer to Weakness 3 (“What I believe the authors should pursue [...]”):**
>
> We thank the reviewer for this insightful suggestion, which we find genuinely exciting. We agree that moving towards top-down alignment of prototypes with human-relevant concepts represents a natural next step for this research. ProtoT definitely has the potential for top-down concept alignment, one possible way consists in integrating auxiliary contrastive objectives that would allow to shape the concepts in the prototypes, pressuring the architecture to lean toward a desired representation. We will include a discussion of this direction in the future work section of the paper, framing it as a promising path towards models whose internal structure can be aligned with stakeholder-relevant concepts.

---

> > ### Author Rebuttal · Reviewer_QVR2 · 2026-04-03
> >
> > I appreciate the answer. I believe the authors should provide a better framing of "interpretable-by-design" and discuss the limitations of the architecture’s interpretability. I have decided to keep my score as is.

---

> > > ### Author Response · Authors · 2026-04-08
> > >
> > > We thank the reviewer for considering our responses and for the helpful suggestions.
> > >
> > > We will update our paper to be more precise in our framing of “interpretable-by-design”, to communicate the architectural origins of prototype interpretability more clearly and with greater nuance, and to add the discussion and new interpretability results referenced in our response to Weakness 1 above, including the comparison with SAE features from LLaMA. In summary, ProtoT does not provide full interpretability, but its design naturally encourages more disentangled representations than standard Transformer-based architectures, with prototypes that can be inspected directly through their highest-activating contexts and often exhibit coherent semantic themes, without requiring auxiliary post-hoc interpretability methods.
> > >
> > > We will also add a short Limitations paragraph clarifying the boundaries of the architecture’s interpretability. In particular, we will clarify that ProtoT does not provide full end-to-end transparency into the model’s internal computation, and that semantic interpretations of individual prototypes remain approximate, including when supported by LLM-assisted labeling.

---

### Official Review · Reviewer_qF4q · 2026-03-12

**Soundness:** 3
**Presentation:** 3
**Significance:** 3
**Originality:** 3
**Overall Recommendation:** 4
**Confidence:** 3

**Summary:**

The paper proposes Prototype Transformer, a LM architecture that replaces self-attention with prototypes. These prototypes are present in every layer and interact with the input through read/write gates and a prefix-memory mechanism. The main idea is have interpretability by design, instead of relying only on post-hoc analysis.

I think the paper is interesting. The direction is novel. The authors also run a broad set of experiments, not just a few visualizations. At the same time, I am not fully convinced by the practical case. ProtoT is still behind LLaMA in perplexity and GLUE, so at this point it looks more like an interesting tradeoff than a strong alternative to standard architectures.

My main concern is the interpretability claim. The paper shows that prototypes can often be given meaningful labels, but this is not the same as showing that the model is more useful in practice for causal analysis or editing. The intervention results are promising, but still limited.

**Compliance With Llm Reviewing Policy:**

Affirmed.

**Final Justification:**

Please see my comments in the Rebuttal Acknowledgment section.

**Key Questions For Authors:**

- How are the prototypes labelled?
- If I wanted to edit a specific behavior out of the model (e.g., remove toxic outputs), how would targeting a prototype in ProtoT compare to applying a method like ROME or steering vectors to LLaMA? Does the inherent design of ProtoT make the edits cleaner or less prone to collateral damage?

**Limitations:**

No.
The paper should discuss at least  more clearly the risk of over-trusting interpretable prototypes, the fact that prototype labels can still be noisy or over-interpreted, and the practical limitation that the model is still weaker/slower than standard LMs. A short explicit limitations paragraph would be enough.

**Strengths And Weaknesses:**

Strengths
- Building interpretability into the architecture is a good direction, and this paper does something real here, not just a small modification
- The paper is experimentally broad. It compares ProtoT with LLaMA, Mamba, and DeltaNet on language modeling, GLUE, throughput, robustness, and interpretability
- The paper is reasonably honest about speed. It does not hide that ProtoT is slower than optimized LLaMA at short context lengths

Weaknesses
- ProtoT is still clearly behind LLaMA, so the paper does not yet show a competitive replacement for standard LMs.
- The linear-complexity claim is fine in theory, but less convincing in practice as the model is still much slower than LLaMA at short context during training. this should be discussed more clearly
- As per interpretability, showing that prototypes can be named is useful, but it is not enough to prove that ProtoT is better than post-hoc methods on normal LMs
- The intervention experiments are limited. They suggest that prototype editing may work, but they do not really show that it is better or cleaner than standard editing methods
- The paper also does not fully answer a practical question: why should someone train ProtoT from scratch instead of taking a strong existing LM and using patching, steering, or editing methods on top of it?

---

> ### Author Rebuttal · Authors · 2026-03-31
>
> We thank the reviewer for their time and suggestions!
>
> ---
>
> > **(from Summary) “[...] interpretability [...] is not the same as [...] causal analysis [...]”**
>
> ProtoT provides explicit intervention sites that make targeted causal analysis of internal model behaviour more direct. The intervention results show that ablating a specific prototype leads to consistent directional changes in the probability of the associated concept, suggesting a functional involvement of the prototype in the model’s computation. Future work may look at more systematic causal analysis.
>
> ---
>
> > **”ProtoT is still clearly behind LLaMA [...]”**
>
> This is not quite true – please see our response to reviewer T9md in ”Answer to Cons 1.”.
>
> ---
>
> > **“The linear-complexity claim [...]”**
>
> The linear-complexity of ProtoT is evident as it outspeeds LLaMA beyond ~32k context (Appendix A.5.1, Table 12). Currently implemented, ProtoT is slower than LLaMA at short sequence lengths – partially because it lacks custom kernels. Indeed, Appendix A.5 (Table 11) shows that ProtoT is a lot slower than LLaMA at 256 context, even though it has significantly fewer FLOPs.
>
> ---
>
> > **“[...] showing that prototypes can be named [...] is not enough to prove that ProtoT is better than post-hoc methods on normal LMs”**
>
> We excluded post-hoc methods because:
>
> 1. Our results show that ProtoT produces interpretable concepts without needing the extra compute and effort necessary for post-hoc methods, unlike LLaMA.
>
> 2. Post-hoc methods can also be applied to ProtoT, hence we see them as complementary approaches.
>
> Regardless, we trained a sparse autoencoder on residual stream of each LLaMA layer and evaluated feature interpretability using LLM-aided scoring pipeline. The results (in response to reviewer T9md Cons 2. and Question 1) show ProtoT leading across all scoring dimensions.
>
> ---
>
> > **“The intervention experiments are limited [...]”**
> >
> > **“The paper also does not fully answer a practical question [...]”**
> >
> > **Question 2: “If I wanted to edit a specific behavior out of the model [...]”**
>
> Standard models like LLaMA are black boxes with entangled, polysemantic features. As highlighted by recent studies [1,2], applying post-hoc patching, steering, or ROME to these dense models often behaves like whack-a-mole: modifying one behavior frequently disrupts numerical stability and degrades the model (PPL degradation, fluency loss).
>
> In contrast, ProtoT has design choices inducing prototype monosemanticity (see response to Reviewer 6GP2 Weakness 2), at levels competitive with Sparse Autoencoders, but at no extra cost (see new experiment in response to reviewer T9md Cons 2. and Question 1). Beyond interpretability, its discrete prototype structure opens natural avenues for steering, editing, and alignment (a promising application highlighted by reviewer QVR2 Weakness 3).
>
> To evaluate how ProtoT compares, we measured the model’s PPL before and after masking specific prototypes. We found < 1% degradation in overall PPL across all interventions. For instance, when predicting the target 'women', disrupting the 'female' prototype (L9 P7) decreased its probability by 17.8% but altered PPL by only +0.4%, and masking the 'male' prototype (L9 P18) increased the target probability by 11.5% with insignificant PPL change (-0.3%). We will update tables 4-7 with the full PPL deltas.
>
> *Limitations:* Our current masking intervention works best when a clear prototype already exists for the targeted concept. For complex concepts not captured by a single prototype, standard continuous steering methods may be more flexible.
>
> *Future work:* We plan to apply steering vectors or ROME-derived methods to a combination of prototype activations rather than MLP weights, which would combine the flexibility of steering with the structural interpretability of ProtoT.
>
> [1] Yang et al. "The butterfly effect of model editing: Few edits can trigger large language models collapse." Findings of ACL 2024.
>
> [2] Vu and Nguyen. "Angular Steering: Behavior Control via Rotation in Activation Space." NeurIPS 2025.
>
> ---
>
> > **Question 1: “How are the prototypes labelled?”**
>
> Labels are assigned via LLM-scoring pipeline: the LLM is prompted to produce a one-line description of each prototype's main theme alongside its scores, including disentanglement and coverage across potentially multiple themes. An example is in Appx. A.8, Figure 15.
>
> ---
>
> > **Answer to Limitations:**
>
> We agree that prototype interpretability should not be over-trusted: while the LLM-scoring reveals high disentanglement for most prototypes, some remain polysemantic, and prototype labels should be treated as approximations rather than ground truth. However, features identified by post-hoc methods on Transformers have similar risks of being noisy and over-interpreted.
>
> “the model is still weaker/slower than standard LMs” - this is not quite true, as discussed above.
>
> We will add a limitations paragraph addressing the nuances of these points.

---

> > ### Author Rebuttal · Reviewer_qF4q · 2026-04-02
> >
> > Thank you for the rebuttal. The extra clarification on prototype labelling and the additional PPL results are useful, and they can improve the paper.
> > However, for me they do not substantially change the overall picture. The main issue is still the practical comparison with strong standard LMs. The response on editing is promising, but still indirect: it argues that editing dense models can be unstable, and shows limited collateral damage for prototype masking, but it does not directly compare against methods like ROME or steering on a competitive baseline. Because of this, the practical case for training ProtoT from scratch remains only partially convincing to me.
> > For this reason, I think my weak accept score is still appropriate. The paper is interesting and I think the direction is valuable, but some of my main reservations remain.

---

> > > ### Author Response · Authors · 2026-04-08
> > >
> > > We thank the reviewer for considering our responses and for the additional comments. We understand that the remaining concern is the practical comparison with strong standard LMs (represented here by LLaMA). As such, further to the points we addressed above, we would like to summarize the performance and efficiency aspects of this comparison more clearly.
> > >
> > > Our experiments show that while ProtoT trails behind LLaMA in terms of perplexity and downstream performance, it outperforms LLaMA in terms of language generation quality. Furthermore, LLaMA’s computation cost scales quadratically in sequence length, both theoretically and in practice, whereas ProtoT scales linearly. This gives ProtoT computational cost and speed advantages at large scale.
> > >
> > > We also thank the reviewer for the additional clarification. We agree that a direct comparison between ProtoT’s native prototype-level interventions and methods such as ROME or steering applied to LLaMA would be a valuable additional experiment for assessing practical trade-offs in edit effectiveness and collateral damage.
> > >
> > > At the same time, we believe the current evidence already supports the paper’s main contribution. Our results show that ProtoT exposes native, directly inspectable intervention sites, and that masking targeted prototypes induces consistent directional changes in the associated concept while producing only minimal change in global PPL. We view this as concrete evidence that ProtoT provides a practically meaningful intervention interface by design, and that these prototypes play a functional role in model prediction rather than merely correlating with the associated concept.
> > >
> > > We will make this framing more explicit in the paper, and we will further highlight that the main contribution is the architecture: a linear-compute language model with strong interpretability properties, where prototypes emerge during standard LM training as directly usable interpretable units. In our experiments, these units achieve stronger interpretability metrics than LLaMA attention heads and sparse autoencoder features, without requiring auxiliary post-hoc interpretability methods, and while maintaining competitive performance across the evaluated baselines. We will also discuss direct comparison with ROME/steering-style methods as an important direction for future work.

---

### Official Review · Reviewer_6GP2 · 2026-03-12

**Soundness:** 3
**Presentation:** 2
**Significance:** 3
**Originality:** 3
**Overall Recommendation:** 5
**Confidence:** 3

**Summary:**

This work focuses on improving the interpretability of generative architectures. Low interpretability is a general problem with DNNs, but is especially impactful for generative architectures given their widespread adoption. As such, the Prototype Transformer (ProtoT) is proposed that replaces the self-attention operator with a linear complexity alternative that essentially aggregates time-based information into constant-sized "prototypes." These learnable vectors capture concrete and sometimes abstract concepts in a demonstrable manner. The proposed architecture scales well in terms of computation and performs fairly on downstream language modeling tasks (e.g. GLUE).

**Compliance With Llm Reviewing Policy:**

Affirmed.

**Final Justification:**

No additional comments beyond what have been mentioned in other parts of the review.

**Key Questions For Authors:**

Portions of Appendix 7 should probably be better integrated into the main body of the work. For a paper like this, you have to firmly establish that the baseline is **not** as capable on your primary, motivating metric (in this case interpretability) as your proposed approach.

Initially, I had a hard time following Table 1, although I understood it after a more careful read. I expect other readers may run into the same issue and perhaps a revision pass could focus on this spot.

**Limitations:**

yes

**Strengths And Weaknesses:**

Strengths:
1. Low interpretability is a huge problem in certain applications. Anything involving generative AI in environments where failure or inexplicable behavior can result in catastrophic consequences essentially requires interpretability (e.g. medicine, high risk manufacturing, defense applications, etc.). Given that, what this work is focused on is clearly important.
2. The work is interesting and is a departure from other efficient attention mechanisms I am familiar with. With respect to that, it appears somewhat unique, even if the idea of "aggregation for compression purposes" is not innately unique.
3. Interpretability claims seem to be backed up by the observed results.

Weaknesses:
1. The model accuracy-to-efficiency tradeoff does not appear to be particularly compelling. Of course, this is offset to a degree by the interpretability improvements. Training ProtoT is slower than LLaMa, although it is significantly faster than Mamba and DeltaNet. Inference does not look very competitive, especially at large sequence lengths, compared to DeltaNet. The test set perplexity results are also somewhat mixed along these lines in unexpected ways (e.g. DeltaNet and Mamba performing notably better in some cases).
2. It isn't clear to me why prototypes and aggregation along those lines is theoretically more interpretable than attention matrices for individual attention heads. This is to say, why can an attention head **not** learn the contextual relationships and ideas surrounding the word "woman" but a prototype can? Do attention heads just have a harder time learning these ideas, if they can learn it? Some more time spent on this would be very useful. Appendix 7 was helpful to this end in terms of establishing empirically that attention heads exhibit less interpretable behavior, but a more comprehensive explanation would be very helpful to establishing the competitive edge ProtoT has.

---

> ### Author Rebuttal · Authors · 2026-03-31
>
> We thank the reviewer for their time and suggestions!
>
> ---
>
> > **Weakness 1: “The model accuracy-to-efficiency tradeoff does not appear to be particularly compelling [...]”**
>
> We admit that ProtoT does not particularly stand out as the best model in terms of accuracy-to-efficiency tradeoff, as compared to the linear-compute baselines. However, some of the speed issues are because we have not developed custom kernels for ProtoT. Also, training ProtoT is only slower than LLaMA at short sequence lengths, and is much faster at large lengths (Appendix A.5.1 (Table 12)) due to its linear compute. Inference may not be very competitive with DeltaNet at long sequence lengths, however, ProtoT has better task performance than DeltaNet at context size 256 (with default model size) and at 512 (with large model size), and we expect ProtoT to keep outperforming DeltaNet as context size is increased in conjunction with model size, as suggested by our discussion in “Long-context scalability” (lines 245-264).
>
> “The test set perplexity results are also somewhat mixed along these lines in unexpected ways (e.g. DeltaNet and Mamba performing notably better in some cases)” – the test-set perplexity results consist of two experiments: large-scale training and long-context scalability. In large-scale training, ProtoT is between Mamba and DeltaNet, which is consistent with the downstream performance results and the text generation quality results. The long-context scalability results do deviate from this, as ProtoT scales worse than the baselines, but in that experiment we also discussed and provided an explanation with empirical justification why that may be, and why it may not matter in practice. In short, the results suggest that the hidden size of ProtoT restricts its long-context scalability the most, and that when jointly scaling context, model size, and data size (in large-scale experiment), ProtoT shows competitive scalability, suggesting that in practice long-context scalability may not be an issue.
>
> ---
>
> > **Weakness 2: ”It isn't clear to me why prototypes and aggregation along those lines is theoretically more interpretable than attention matrices for individual attention heads [...]”**
>
> We wholeheartedly thank the reviewer for pushing us to explain this better and compare it with self-attention! The explanation on lines 139-145 (left) of the paper is only part of the story. Below is a more detailed and accurate one, which we will update in the paper. As a reminder, on a high level (simplified), ProtoT works like this at the write gate (the gate at which we interpret it): it has R communication channels, each defined by a prototype (fixed-size parameter vector); information is aggregated via a weighted average of the values up to the current token position, where the weights are obtained via dot-product between the prototype and the states (“keys”). Now the explanation:
>
> - For ProtoT, each prototype is determining the weights by which the context is being averaged in its corresponding channel. This averaging makes multiple meanings combine into a compromise representation, blurring the individual meanings and creating competitive pressure, which pushes the model to use separate channels for them. Whereas, for self-attention, the query at the given position is determining the weights by which the context is being averaged for that head. And as such, the query can vary across the sequence positions, forming different retrieval patterns, thus it can become polysemantic across the sequence length (vs ProtoT’s monosemanticity across sequence length).
>
> In summary, it is not impossible for attention heads to learn the concept of ‘woman’ but there is no pressure towards it and polysemanticity in attention heads becomes a natural and effective strategy for learning large datasets. In ProtoT we have the opposite, the architecture is pressured towards concept disentanglement and the emerging representation can be directly inspected for interpretability, without the necessity of post-hoc methods.
>
> ---
>
> > **Question 1: “Portions of Appendix 7 should probably be better integrated [...]”**
>
> Thank you for the suggestion, we agree that more space should be dedicated to establishing that ProtoT is more capable than LLama on interpretability. We will integrate the important points of Appendix 7 in the main body.
>
> To better prove ProtoT advantages on interpretability we designed an additional LLM-aided scoring experiment, adding a comparison with LLaMA with Sparse Autoencoder (SAE), showing that ProtoT scores better across all scoring dimensions. Additional details and discussion can be found in our answer to reviewer T9md (in “Combined answer to [...]”).
>
> ---
>
> > **Question 2: “Initially, I had a hard time following Table 1 [...]”**
>
> Thank you for the suggestion. We will make this clearer by splitting the table into two tables – one addressing the large-scale training and one addressing the cong-context scalability experiment.

---

> > ### Author Rebuttal · Reviewer_6GP2 · 2026-04-03
> >
> > Thank the authors for the rebuttal.

---

> > > ### Author Response · Authors · 2026-04-08
> > >
> > > We thank the reviewer for the positive feedback and for their time and consideration.

---

### Official Review · Reviewer_T9md · 2026-03-13

**Soundness:** 2
**Presentation:** 3
**Significance:** 2
**Originality:** 3
**Overall Recommendation:** 4
**Confidence:** 3

**Summary:**

This paper introduces the Prototype Transformer, an autoregressive language model architecture that replaces standard self-attention with a prototype-based mixer module. Each layer uses R learnable parameter vectors (prototypes) that define independent communication channels: a write gate routes sequence information into channels via prototype-input similarity, a time-discounted prefix mean aggregates past context per channel, and a read gate retrieves information back into the sequence. This yields linear complexity in sequence length. The authors train ProtoT on FineWeb-Edu and compare against LLaMA, Mamba, and DeltaNet. They show that prototypes automatically learn disentangled, nameable concepts (e.g., "woman," "COVID," dates, punctuation) verified through LLM-aided scoring, demonstrate targeted behavior editing via prototype intervention, report competitive GLUE and text generation performance, and analyze robustness to meaning-preserving perturbations.

**Compliance With Llm Reviewing Policy:**

Affirmed.

**Final Justification:**

The rebuttal addressed my main concern.

**Key Questions For Authors:**

What disentanglement and coverage scores does the LLM judge assign to random subsets of 10 sentences drawn from FineWeb?

**Limitations:**

No limitations, please see my comments in the weakness

**Strengths And Weaknesses:**

Pros:
1. The paper is easy to follow and well organized.
2. The prototype-based design is a genuinely creative approach to building interpretability into the architecture itself rather than relying on post-hoc analysis, and the evidence that prototypes learn disentangled nameable concepts without explicit supervision is compelling.

Cons:
1. My main concern is the performance gap with LLaMA is substantial and under-acknowledged. At large scale, ProtoT achieves 29.5 perplexity versus LLaMA's 25.8: a 14.3% gap that is quite large in absolute terms for language modeling. On GLUE, ProtoT trails LLaMA by 4 points overall (67.6 vs 71.6) and badly on specific tasks like STS-B (66.2 vs 78.3) and CoLA (27.7 vs 31.5). The paper frames this as "competitive" and "close to state-of-the-art," but this gap needs more honest treatment. The argument that "this is the first iteration" is not a sufficient justification.

2. The LLM-aided scoring (using GPT-5.1 to judge disentanglement) is circular in a subtle way: the judge is asked to find themes in top-activating sequences, but almost any subset of natural language sentences will share some nameable theme if you look hard enough. There's no null-model baseline : what scores would random subsets of sentences get? Without this, high disentanglement scores are hard to interpret. The comparison to LLaMA attention heads (Figure 24) is also not entirely fair, since attention heads and prototypes are architecturally different objects with different numbers and roles. A fairer comparison might use SAE features from LLaMA, which are specifically designed to extract monosemantic directions.

---

> ### Author Rebuttal · Authors · 2026-03-31
>
> We thank the reviewer for their time and suggestions!
>
> ---
>
> **Answer to Cons 1. (“My main concern is the performance gap with LLaMA is substantial and under-acknowledged [...]”):**
>
> Thank you for your comment - we will acknowledge more prominently in the paper the nuances of the performance gap between ProtoT and LLaMA.
>
> 1. While ProtoT trails behind LLaMA in terms of perplexity and downstream performance, it is worth noting that ProtoT outperforms LLaMA in terms of language generation quality (Appendix B and lines 265-274 (left) in the paper, with updated results below).
>
> 2. More importantly, LLaMA’s computation cost scales quadratically in sequence length, whereas ProtoT scales linearly. This means the two models are not really in the same class, and the linear baselines (Mamba and DeltaNet) are more suitable to compare against. And indeed, the results show that ProtoT’s performance is consistently between that of Mamba and DeltaNet. These points may not have been communicated very well in the paper, which we will improve in the next revision.
>
> 3. We have updated the text-generation experiment with more prompts, for better confidence in the results. We now use 100 rather than 20 prompts (so 100x6 pairs of comparisons for the LLM judge), randomly chosen from the FineWeb-test set. The results in the table below show a much narrower Elo gap with Mamba (nearly identical now, while previously Mamba seemed significantly better), while still outperforming LLaMA and DeltaNet.
>
> **Table: Text-generation experiment**
>
> | **Method** | **Final Elo ↑** |
> |---|---:|
> | Mamba | **1041.79** |
> | ProtoT | 1021.24 |
> | LLaMA | 975.18 |
> | DeltaNet | 961.80 |
>
> ---
>
> **Combined answer to Cons 2. and Question (1/1):**
>
> We thank the reviewer for their comments, which we address by designing a new experiment (below) including the requested null-model baseline and SAE features from LLaMA.
>
> Regarding the comparison with LLaMA attention heads, we agree that SAE features represent a stronger and fairer baseline, as they are specifically designed to extract monosemantic directions. At the same time, we believe the attention head comparison is still valuable: it demonstrates that interpretable concept representations do not emerge natively in standard transformer architectures without costly post-hoc methods, whereas ProtoT learns them for free during standard language modeling training. These are complementary points rather than competing ones. Furthermore, post-hoc interpretability methods such as SAEs can, in principle, also be applied to the ProtoT architecture.
>
> *Experimental setup:*
>
> - *SAE comparison:* We train a sparse autoencoder on the residual stream of each layer of the trained LLaMA model, using 100M tokens and a dictionary size of 4096. We select 32 features per layer (to match the number of prototypes) using two methods: (1) highest activation variance: the features that fire most selectively and are therefore the most probable carriers of disentangled concepts; (2) highest activation frequency: the features that fire most often and are thus best covered by the validation data. Note that this design purposely favors SAE as we select a cherry picked subset of the most relevant features, while in ProtoT we select all prototypes indiscriminately. We identify the 10 most activating sentences per feature and then apply the LLM-aided scoring pipeline.
>
> - *Null model comparison:* We select a number of random subsets of 10 sentences from Fineweb equal to the number of prototypes (32 prototypes per 12 layers = 384 total prototypes) and apply the LLM-aided scoring pipeline.
>
> *Results:*
>
> | **Method** | **Disentanglement ↑** | **Coverage ↑** | **Num. Themes ↓** |
> |---|---:|---:|---:|
> | ProtoT | **6.52 ± 1.93** | **7.88 ± 2.25** | **3.86 ± 1.94** |
> | LLaMA SAE – Top Variance | 5.91 ± 1.92 | 7.86 ± 1.84 | 4.33 ± 1.82 |
> | LLaMA SAE – Top Frequency | 5.52 ± 1.81 | 7.47 ± 1.88 | 4.68 ± 1.82 |
> | LLaMA Attention Heads | 5.02 ± 1.28 | 6.69 ± 1.84 | 5.02 ± 1.55 |
> | Null Model | 3.20 ± 1.02 | 4.03 ± 1.78 | 6.97 ± 1.28 |
>
> The results (in the table above) show that while SAE features score substantially above the null model and above raw LLaMA attention heads, ProtoT still leads across all scoring dimensions. Furthermore, the null-model baseline is definitely worse than the other models, validating the effectiveness of LLM-aided scoring.

---

> > ### Author Rebuttal · Reviewer_T9md · 2026-04-03
> >
> > Thanks for your rebuttal. Please include these new results in the revised paper, and I will update my score accordingly.

---

> > > ### Author Response · Authors · 2026-04-08
> > >
> > > We thank the reviewer for the positive feedback and for their willingness to update their score accordingly. We will certainly include these new results in the revised version of the paper. As a procedural note, ICML does not allow authors to upload a revised manuscript during the author-feedback period, so for now we have provided these results in the discussion above for the reviewer’s consideration.

---

### Decision · Program_Chairs · 2026-04-30

**Decision:**

Accept (regular)

**Comment:**

The paper introduces the Prototype Transformer, an autoregressive language model that replaces traditional self-attention with a linear-complexity mechanism. Although the design theoretically isolates distinct concepts into discrete communication channels, the reviewers (e.g. 6GP2, QVR2) found the accompanying empirical claims heavily overstated. Despite the positive score, I tend to agree that the camera-ready version needs to adjust the language and the claims and position them appropriately with respect to the literature. In addition, there seem to be less related work referenced, while there are many (posthoc) interpretability methods, so the authors are encouraged to include such methods.  Overall, it seems an interesting idea and the execution seems good, so I will recommend acceptance.